# KAP1 is an antiparallel dimer with a functional asymmetry

Giulia Fonti[1,2], Maria J Marcaida[1,2], Louise C Bryan[3], Sylvain Träger[1,2], Alexandra S Kalantzi[1,2], Pierre-Yves JL Helleboid[4], Davide Demurtas[5], Mark D Tully[6], Sergei Grudinin[7], Didier Trono[4], Beat Fierz[3], Matteo Dal Peraro[1,2]

**KAP1 (KRAB domain–associated protein 1) plays a fundamental role in regulating gene expression in mammalian cells by recruiting different transcription factors and altering the chromatin state. In doing so, KAP1 acts both as a platform for macromolecular interactions and as an E3 small ubiquitin modifier ligase. This work sheds light on the overall organization of the full-length protein combining solution scattering data, integrative modeling, and single-molecule experiments. We show that KAP1 is an elongated antiparallel dimer with an asymmetry at the C-terminal domains. This conformation is consistent with the finding that the Really Interesting New Gene (RING) domain contributes to KAP1 auto-SUMOylation. Importantly, this intrinsic asymmetry has key functional implications for the KAP1 network of interactions, as the heterochromatin protein 1 (HP1) occupies only one of the two putative HP1 binding sites on the KAP1 dimer, resulting in an unexpected stoichiometry, even in the context of chromatin fibers.**

## Introduction

KAP1—KRAB (Krüppel-associated box) domain–associated protein 1—also known as TIF1$\beta$ (transcription intermediary factor 1$\beta$) or TRIM28 (tripartite motif containing protein 28) is a central regulator that controls the fate of the genetic material by recruiting transcription factors and altering the chromatin environment (1, 2). KAP1 is, thus, essential for early development (3) and has been linked to fundamental cellular processes such as differentiation (4, 5), gene silencing (6, 7, 8, 9), transcription regulation (10, 11, 12, 13), and DNA damage response (8, 14, 15, 16, 17, 18, 19). Moreover, its involvement in control of behavioral stress and tumorigenesis makes it an attractive therapeutic target (20, 21, 22, 23, 24, 25, 26, 27).

KAP1 belongs to the superfamily of the tripartite motif-containing (TRIM) proteins that includes more than 60 members in humans with variable C-terminal domains (28). The TRIM family is defined by the presence of a highly conserved N-terminal domain consisting of a Really Interesting New Gene (RING) finger domain, one or two B-box domains ($B_1$ and $B_2$), and a long coiled coil (CC), collectively called RBCC (28) (Fig 1A). The RING domain contains a regular arrangement of cysteine and histidine residues that coordinate two zinc ions tetrahedrally in a unique "cross-brace" fold and acts as an E3 SUMO (small ubiquitin modifier) and E3 Ubiquitin ligase (29, 30, 31). The B-box domain shares the RING domain fold and may bind one or two zinc ions (33, 34). The CC of KAP1 is estimated to be very long (~200 Å) and together with the $B_2$ is likely used to mediate protein–protein interactions (35).

KAP1 is a member of the TRIM C-VI subfamily, together with TRIM24 and TRIM33, characterized by the presence of a tandem plant homeodomain (PHD) and bromodomain (Br) typically involved in the recognition of various histones modifications (37, 38). However, the C-terminal tandem PHD-Br domain of KAP1 shows a unique function acting as an E3 SUMO ligase, promoting both the auto-SUMOylation of the protein (39) and the SUMOylation of other substrates (40, 41). The NMR structure of the KAP1 PHD-Br domain elucidated how the two domains cooperate as one E3 SUMO ligase unit (36). The auto-SUMOylation of the C-terminal PHD-Br domain is necessary for the binding of KAP1 to the chromatin-remodeling enzymes such as the methyltransferase SET domain bifurcated 1 (SETDB1), the nucleosome remodeling and histone deacetylation complex, the histone deacetylase, the nuclear co-repressor, inducing the deposition of post-translational modifications (PTMs), and consequently the creation of an heterochromatin environment (42, 43, 44, 45, 46) (Fig 1A).

The N-terminal domain and the C-terminal PHD-Br domain are connected by a long loop of ~200 amino acids without any predicted structure, where the heterochromatin protein 1 (HP1) is recruited by KAP1 on a specific binding domain (HP1BD, Fig 1A) (47, 48). HP1 is a transcriptional repressor that directly mediates the formation of higher order chromatin structures (49, 50), characterized by the presence of an N-terminal chromo domain (CD) and a C-terminal chromo shadow domain (CSD), linked by an unstructured hinge region. The CD binds directly the methylated lysine 9 of histone H3, whereas the CSD forms a symmetrical dimer that mediates interactions with other proteins by recognizing a PXVXL penta-peptide

¹Institute of Bioengineering, School of Life Sciences, Ecole Polytechnique Fédérale de Lausanne, Lausanne, Switzerland   ²Swiss Institute of Bioinformatics, Lausanne, Switzerland   ³Institute of Chemical Sciences and Engineering, School of Basic Sciences, Ecole Polytechnique Fédérale de Lausanne, Lausanne, Switzerland ⁴Global Health Institute, School of Life Sciences, Ecole Polytechnique Fédérale de Lausanne, Lausanne, Switzerland   ⁵Interdisciplinary Centre for Electron Microscopy, Ecole Polytechnique Fédérale de Lausanne, Lausanne, Switzerland   ⁶European Synchrotron Radiation Facility, Grenoble, France   ⁷University Grenoble Alpes, Centre National de la Recherche Scientifique, Inria, Grenoble Institut Polytechnique de Grenoble, Laboratoire Jean Kuntzmann, Grenoble, France

Correspondence: matteo.dalperaro@epfl.ch; maria.marcaidalopez@epfl.ch

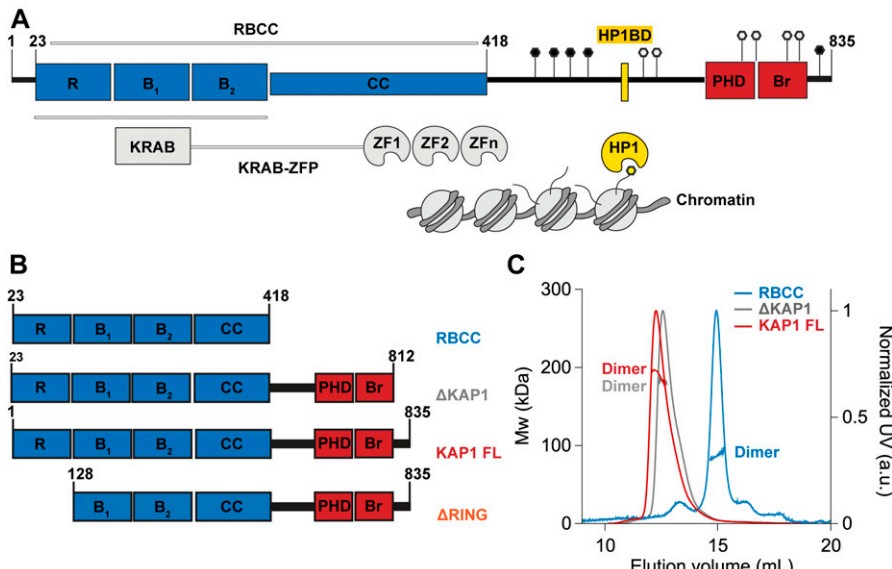

**Figure 1. Oligomerization state of KAP1.**
**(A)** KAP1 sequence architecture. The different KAP1 domains are reported on top. The structures of the individual RING, B-box 1, and B-box 2 domains as well as the PHD-Br domain have been solved by X-ray crystallography and NMR, respectively (PDB IDs 6I9H (29), 6O5K (33), 2YVR, and 2RO1 (36)). Key residues affected by PTM are also highlighted (phosphorylation sites in black and SUMOylation in white). **(B)** Schematic of the different constructs used in this study. **(C)** SEC-MALS analyses of the KAP1 constructs show that they are all dimers. The traces are colored according to the KAP1 construct and show the normalized elution profile measured at 280 nm (right axis) and the calculated molecular weight of the selected peaks in kDa (left axis): 88 kDa for the RBCC domain, 183 kDa for ΔKAP1, and 190 kDa for KAP1 FL

motif (where X is any amino acid) (51). The interaction with HP1 is crucial for KAP1 to regulate the chromatin state (47), carry out its gene silencing function (52), and respond to DNA damage (17).

KAP1 mediates gene silencing of transposable elements by recruiting KRAB-zinc finger proteins (KRAB-ZFPs), which constitute the largest family of TFs with more than 350 members and which are present only in tetrapod vertebrates (53, 54). Their function is mainly unknown, but some members have been involved in diverse processes such as embryonic development, tissue-specific gene expression, and cancer progression (55). KRAB-ZFPs recruit KAP1 to specific genetic loci, via the interaction between the RBCC and the KRAB repression module (9, 35, 55), in turn, KAP1 recruits chromatin remodeling enzymes to tri-methylated lysine 9 of histone H3. This modification, along with the DNA compaction caused by HP1 binding, creates a heterochromatin environment silencing gene expression (7).

Despite its emerging fundamental importance, KAP1 and its interaction with binding partners are still poorly characterized from a structural standpoint. Crystal structures of the CC with or without additional B-box or C-terminal domains have been solved for some TRIM family members (TRIM25 (56, 57), TRIM5 (58, 59), TRIM20 (60), and TRIM69 (61)), showing that the RBCC is an elongated domain with an antiparallel homodimer conformation. However, there is insufficient structural information about full-length proteins of the TRIM family. Here, we present the overall structural organization of human full-length KAP1 in solution by integrating biochemical characterization, small-angle X-ray scattering data, molecular modeling, and single-molecule (sm) experiments. We demonstrate that KAP1 is an elongated antiparallel dimer in which the C-terminal domains are generating an asymmetrical architecture. The observation that the N-terminal RING domain is necessary for auto-SUMOylation is consistent with this conformation of the FL KAP1. Furthermore, we show that an asymmetric organization is manifest also when KAP1 is in complex with HP1, which occupies only one of the two existing binding sites in the KAP1 dimer. We discuss the

implications of this asymmetry for HP1 binding kinetics in the chromatin context and in general for KAP1 function.

# Results

## KAP1 is a homodimer in solution

One of the main structural features of the TRIM family members characterized so far is their ability to form homodimers through their CC region (56, 57, 58, 60, 62) (Fig 1A). Biochemical and bioinformatics analyses suggest that this organization could be a common feature for all the TRIM family members (57). Although the homodimer state is the preferred one, other studies have reported the possibility for some TRIM family members to form heterodimers or heterotrimers (63). Moreover, another layer of complexity is added by the possibility to form high-order oligomers mediated by the RING and B-box domains (64). Unlike the other TRIM family members, KAP1 has been previously suggested to fold preferentially as a homotrimer alone in solution and in complex with the KRAB domain of KRAB ZFPs (35). Furthermore, hetero-interactions with TRIM24 and further associations as hexamers have been suggested (63). To shed light on the oligomerization state of KAP1, we performed size exclusion chromatography coupled to multiangle light scattering (SEC-MALS) and sedimentation velocity analytical ultracentrifugation (SV-AUC) experiments. Three different constructs (Fig 1B) have been used: (i) the RBCC domain (RBCC, 23–418) mainly responsible for oligomerization, (ii) a KAP1 construct nearly covering the whole sequence (ΔKAP1, 23–812), and (iii) the complete KAP1 full-length protein (KAP1 FL, 1–835). SEC-MALS experiments were performed across a concentration range of 10 to 40 μM (Fig 1C). No concentration dependence in either elution volume or mass estimation was observed in these conditions. The resulting molecular weights (Mw$_s$) were determined to be 88 kDa for the RBCC

domain, 183 kDa for ΔKAP1 and 190 kDa for KAP1 FL, in agreement with dimeric species of expected Mw$_s$ of 92 kDa (RBCC), 175 kDa (ΔKAP1), and 182 kDa (KAP1 FL) (Fig 1C). Under the concentration range tested, both SEC-MALS and SV-AUC (Fig S1) consistently indicated a dimeric conformation for KAP1.

## KAP1 is an antiparallel elongated dimer determined by the CC domain

To gain more information on the structural organization of KAP1 in solution, we performed small-angle X-ray scattering (SAXS). Size-exclusion chromatography was coupled in line with SAXS to avoid aggregation effects during the experiments (Fig 2A). Although SAXS data were recorded for the three KAP1 constructs, ΔKAP1 results showed almost identical as those of KAP1 FL and are reported in Fig S2. The data were collected across a concentration range of 9–15 mg/ml (100–200 μM). The Guinier analysis of the scattering curves showed good linearity, indicating neither aggregation nor polydispersity effects and gave an estimated radius of gyration (R$_g$) of 83 Å for RBCC, 90 Å for KAP1 FL, and 89 Å for ΔKAP1 (Fig S3 and Table S1). Moreover, the value for the cross section R$_g$ (R$_{gc}$, Fig S4) was similar for KAP1 FL (35.8 Å) and ΔKAP1 (38.8 Å), whereas it was smaller for the construct containing only the N-terminal RBCC domain (20.2 Å) (Table S1) pointing to an elongated structure for KAP1. In addition, the SV-AUC experiment on the RBCC construct showed the protein to sediment as a single species with an S$_{20}$,w of 2.3 and a frictional ratio (f/f$_o$, where f$_o$ is the frictional coefficient of a smooth compact sphere) well above 1.3, indicating the elongated nature of the molecule and confirming the SAXS observation (Fig S1). Furthermore, the analysis of the Kratky plot, which can detect features typical of multi domain proteins with flexible linkers, indicates a large flexibility of KAP1 (Fig S5). This observation agrees with the prediction of an unstructured 200-residue-long loop connecting the N-terminal RBCC and the C-terminal PHD-Br domains.

MALS and SAXS analyses reveal that KAP1 is an elongated flexible dimer. Therefore, to gain insight into its mass distribution, we

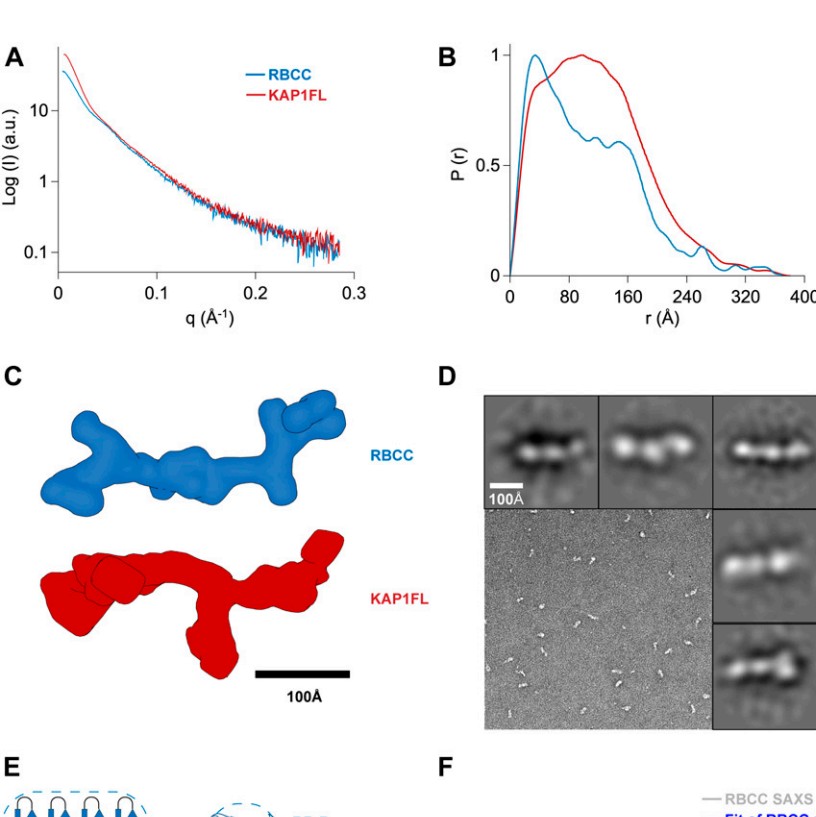

**Figure 2. Overall architecture of KAP1.**
**(A)** SAXS scattering curves. **(B)** Pair-distance distribution functions. **(C)** Ab initio bead models created using GASBOR (65) shown in surface representation. **(D)** Representative TEM image of negatively stained KAP1 FL and gallery of 2D class averages. **(E)** A representative atomic model of the antiparallel RBCC dimer (different protomers are colored in blue and grey) capturing the main features of the ensemble of models able to fit the SAXS data (see Fig S6). The end of the RBCC contains the TIF1 signature sequence (TSS) domain, which is rich in phenylalanine and tryptophan residues (2) and is modeled as a helix that packs against the main long CC domain forming overall a four-helix bundle. **(F)** Fit of representative scattering profiles of the molecular models shown in Fig S6 (in blue) to the SAXS scattering data (in grey) using Pepsi-SAXS (66) (fitting with 0.8 < χ$^2$ < 1).

compared the pair distribution functions, *P(r)*, of the RBCC domain and KAP1 FL. Both show the signature of an elongated molecule with very large maximum dimension ($D_{max}$) values and a main peak at a shorter radius (Fig 2B). The *P(r)* function for the RBCC is clearly bimodal, containing two peaks one centered at 40 Å and the other one at 160 Å with a $D_{max}$ of 370 Å (Fig 2B and Table S1) suggesting a dumbbell-shaped macromolecule with the RBCC domain arranged in an antiparallel fashion separating the two N-terminal RING/B-box 1/B-box 2 ($RB_1B_2$) modules by its length (160 Å). The *P(r)* function of KAP1 FL is also characteristic of an elongated molecule, but surprisingly, it shows a similar $D_{max}$ (380 Å) as for the RBCC domain alone. This observation, together with the increase in $R_{gc}$, leads to the conclusion that the extra domains present in the full-length protein are not fully extended but are in proximity to the N-terminal RBCC domain. This conclusion is also supported by the loss of the bimodality in the *P(r)* function compatible with a more compact protein in which the extra C-terminal domains are close by the RBCC domains (Fig 2B). In addition, ab initio bead models calculated directly from the SAXS scattering curves (Fig 2C) provided conformations of similar length (~320 Å), but differed width, in agreement with the larger $R_{gc}$ of KAP1 FL.

Transmission electron microscopy was used to visualize negatively stained KAP1 FL to independently measure its size and shape (Fig 2D). Rod-like particles with a multidomain organization could be readily observed in the micrograph when sorted into 2D class averages. In agreement with the SAXS data, the particles have a length of around ~280 Å.

The RBCC domain being a dimer with an elongated dumbbell-shaped conformation is only compatible with an antiparallel conformation in which the two tandem $RB_1B_2$ modules are separated by the CC domain (Fig 2E). Therefore, we could build an atomic model of the RBCC domain using homologous TRIM structures as templates. Despite the low-sequence homology among the family, model building was facilitated by conservation of the canonical left-handed CC motif and the zinc-coordinating His/Cys pairs of the B-box 2 domain among 54 human TRIM proteins (57). The RBCC model showed an elongated structure of ~310 Å in length, with the $RB_1B_2$ modules separated by ~160 Å, the dimension of the dimeric antiparallel CC domain (Fig 2E). Furthermore, the sedimentation and diffusion coefficients of this RBCC model were computed using HYDROPRO (67), resulting in values that agree with those extracted from the SV-AUC experiment (Fig S1).

To sample the flexibility of the RBCC domain in solution, an ensemble of RBCC models was created by varying the positions of the RING and B-box 1 domains with respect to the B-box 2 and CC domains (whose position is conserved in several crystal structures). These models were fitted to the SAXS data using a $\chi^2$–minimizing optimization method (68) (see the Materials and Methods section and Fig 2F), resulting in highly similar structures that explore a tight conformational space (Fig S6). Interestingly, when compared with published SAXS data of other TRIM family members, the $R_{gc}$ of the RBCC KAP1 is ~20 Å, remarkably smaller than that of RBCC-TRIM25 (~31 Å) and RBCC-TRIM32 (~32 Å) (62). In the first case, the difference in $R_{gc}$ value can be due to the fact that the $RB_1B_2$ modules of TRIM25 are thought to fold back on the structure of the CC, whereas for KAP1 can be fitted only at the extremes of the CC. In the second case, RBCC-TRIM32 behaves as a tetramer in solution (62), whereas we

found KAP1 exclusively as a dimer in solution at the sampled concentrations (Fig 1C).

## KAP1 dimers are natively asymmetric in solution

The PHD-Br domain of KAP1 acts as an E3 SUMO ligase (39), which renders KAP1 unique within the TRIM family. The overall organization of the full-length KAP1 is unknown as well as the possible interactions, functional cooperation, or structural stabilization between the RBCC and C-terminal domains. Therefore, to gain deeper insight into the domain organization of KAP1, very flexible according to the Kratky plot, we devised an integrative modeling strategy (69, 70, 71) that uses nonlinear Cartesian Normal Mode Analysis (NOLB NMA) (68) to optimally fit the SAXS data.

Briefly, 1,000 randomized atomic models of KAP1 FL were generated based on the dimeric RBCC model (Fig 2E) and the structure of the C-terminal PHD-Br domains, where ~200 amino acids between the RBCC and the PHD-Br domains were modeled as a random coil structure (see the Materials and Methods section). The NMA technique was used to efficiently explore the configurations of the flexible linkers and rigid domains in a reduced conformational space of dimensionality 60 (66). The final models converged to an ensemble of structures well consistent with SAXS data ($\chi^2 = 1.1 \pm 0.3$), reducing their $R_g$ values from ~94 ± 7 to ~88 ± 2 Å, a value similar to the experimental $R_g$ value for KAP1 FL (Figs 3A and S7 and Table S1).

To better characterize the global architecture of the KAP1 FL, the relevant distances of the mutual positions of the domains separated by the flexible loops ($dA_i$ and $dB_i$ in Fig 3B) were used. As expected, the initial models had a very heterogeneous ensemble of conformations; however, after flexible fitting, two distinct clusters could be identified (Fig 3C). Because of the dimeric nature of KAP1, the two resulting clusters are symmetric and they, as well as their centers (white dots in Fig 3C), can be considered equivalent (Figs 3C and S7A). This cluster contains models that are characterized by a large displacement (~120 Å) of one PHD-Br domain from the $RB_1B_2$ region and a small displacement (~60 Å) of the other PHD-Br domain from the remaining $RB_1B_2$ (Figs 3C and S7). This is highlighted in the 1D plot in Fig 3D where the fit models are not randomly distributed according to the distance between the N- and C-terminal domains but are organized as two separated Gaussian distributions centered at ~60 and ~120 Å. The models contained in the first density level best describe the conformation of KAP1 FL in solution (see calculated scattering curves in Fig 3F). An ensemble of some of these models is shown in Fig S7B, whereas the cluster center structure is shown in Fig 3E as representative model able to capture the general features of the ensemble. Within this ensemble, the two PHD-Br domains are never found simultaneously close to the $RB_1B_2$ domains, contrary to what was proposed for the TRIM25 PRYSPRY domain and the NHL repeats of TRIM32 (62). If such conformation is imposed, this induced large distortions of the CC domain and poor data fit ($\chi^2 > 80$). Similarly, flexible fitting never selected conformations where the two C-termini are fully extending away from the RBCC domains ($\chi^2 > 75$). Consistently, *P(r)* functions calculated a posteriori from models optimally compare with experimental *P(r)* only for the cluster center model structures (Fig S8).

Therefore, SAXS data strongly indicate that the organization of KAP1 FL is a natively asymmetric dimer, arranged with the

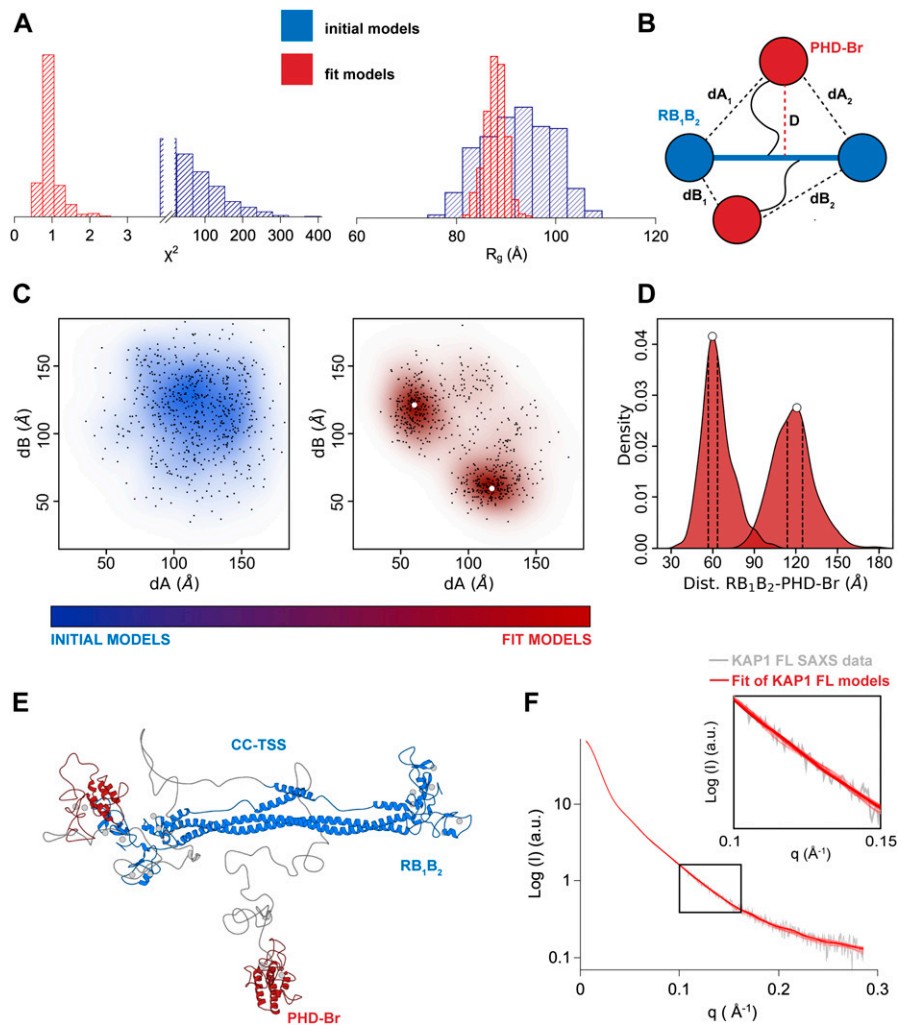

**Figure 3. Native asymmetry of KAP1.**
**(A)** $\chi^2$ (left) and $R_g$ (Å) plots (right) of the initial (blue) versus final (red) model structures, with final distributions centered around $\chi^2 = 1$ and the experimental $R_g$ (90 Å). **(B)** Schematic representation of the KAP1 FL dimer where the distances, $D$ (maximum initial distance), $dA_{1,2}$, and $dB_{1,2}$ are displayed. **(C)** Plot of the initial versus final models according to their $dA$ versus $dB$ distance values. The centers of the clusters are highlighted by white dots. **(D)** 1D density distribution of the clustered models according to the distances between the PHD-Br and the $RB_1B_2$ domains showing two clear Gaussian distributions (see also Fig S7). The dotted lines show the limits of the first density level containing 10% of the models. **(E)** Cartoon representation of the cluster center model of KAP1 FL representative of the ensemble that optimally fits the SAXS data. The ensemble of selected models belonging to the first density level in D is shown in Fig S7B. **(F)** Comparison between the SAXS scattering profile (grey) and the calculated scattering profiles (red) from the models in the first density level using Pepsi-SAXS ($0.8 < \chi^2 < 2$).

C-terminal halves close by the RBCC domain and two distinct distributions of the distances between the PHD-Br and the $RB_1B_2$ domains. The asymmetry seems to originate exclusively from the C-terminal domains, as the ensemble of RBCC structures can be considered symmetric, at least at this level of resolution. Ab initio bead models using the SAXS scattering profiles support in fact this conclusion as we obtained quasi-symmetric structures also without imposing any symmetry (Figs 2C and S6). In the case of FL KAP1, however, imposing twofold symmetry generates a set of bead models with a much divergent range of conformations than that generated without imposing symmetry restraints (Fig 2C).

## KAP1 architecture allows RING-dependent auto-SUMOylation

We asked next if this unexpected conformation of the KAP1 C-terminal domain in solution may have direct implications for its function. The residue C651, located in the PH domain, has been shown to be the key residue for the auto-SUMOylation of the C terminus (39) (Fig 1A). This PTM is fundamental because it allows KAP1 to interact with chromatin remodeling enzymes inducing the formation of heterochromatin (43, 44). However, previous studies

reported that the intact PH domain was necessary for the auto-SUMOylation of many but not all the sites in KAP1 (39), implying the existence of a second catalytic site. Similarly, the RING domain has been shown to strongly interact with the Ubc9 E2 SUMO ligase and be fundamental for the SUMOylation process of KAP1 substrates such as the interferon regulatory factor 7 (IRF7) (31) and the neurodegenerative disease driving proteins τ and α-synuclein (32). These findings as well as the proximity of one of the C-terminal halves to the RBCC proposed by our SAXS-based ensemble (Fig 3E) hint a role for the RING domain in the auto-SUMOylation of KAP1. To prove this hypothesis, we performed a SUMOylation assay in vitro using purified proteins, where KAP1 acts as an E3 SUMO ligase, auto-SUMOylating itself. We compared the auto-SUMOylation of KAP1 FL, RBCC, KAP1 FL C651A mutant, and a deletion mutant missing the N-terminal RING domain (ΔRING) (Fig 1B), after having checked that they were properly folded (Fig S9). As expected, the RBCC domain was not SUMOylated, but differences in the time-dependent SUMOylation of the other variants were observed (Figs 4 and S10). The C651A mutant showed residual SUMOylation activity with respect to the wild-type protein (Fig S10) as previously reported (39). More importantly, we found that also the ΔRING construct presented a much lower auto-SUMOylation

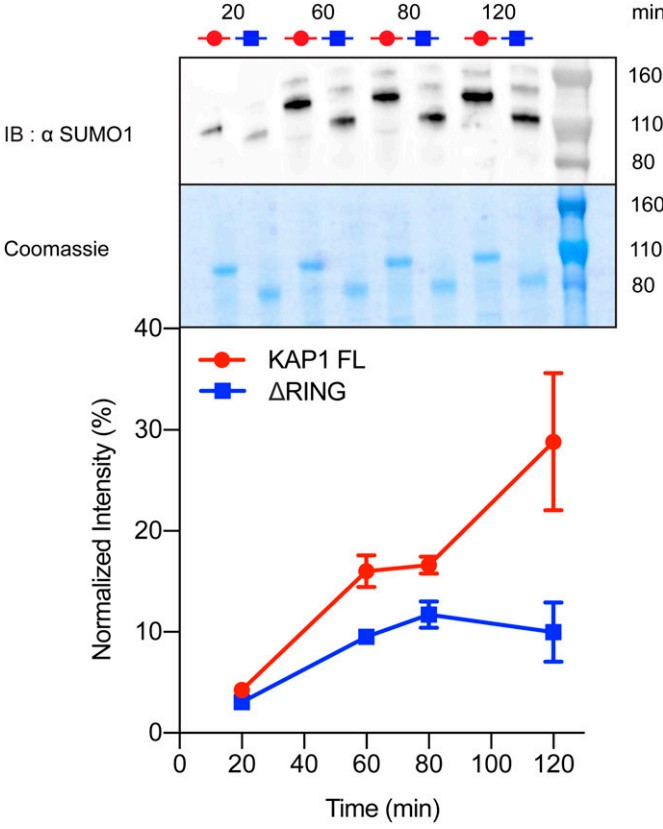

**Figure 4. The RING domain contributes to auto-SUMOylation of KAP1.**
In vitro SUMOylation of KAP1, where reactions containing E1, E2 (Ubc9), SUMO1, and KAP1 variants (FL and ΔRING) were incubated for the time indicated (20 to 120 min). The samples were analyzed by Western blot with anti-SUMO1 antibody (top) and Coomassie blue staining (center). The intensity of the bands was quantified using ImageJ (74) and normalized with respect to the sum of all intensities. The graph shows the mean value and the standard error of the mean for each time step for the three technical replicas (uncropped images shown in Fig S10). Auto-SUMOylation for RBCC, C651A mutant, and controls are also reported in Fig S10.

activity than KAP1 FL (about one-third of FL activity at 120 min, Figs 4 and S10). Taken together, these results show that not only the PHD but also the N-terminal RING domain is implicated in the auto-SUMOylation process of the C-terminal domain. The conformation discovered by our SAXS-based modeling (Fig 3E) is, thus, consistent with a potential communication between the two terminal domains, which can both cooperate in the E3 SUMO ligase activity of KAP1. The RING domain can thus act as a second active site, as it is able to SUMOylate other substrates (31, 32). Interestingly, RING-mediated auto-SUMOylation has been observed also in another TRIM family member, the protein promyelocytic leukemia (TRIM19), which is involved in the formation of nuclear structures called promyelocytic leukemia nuclear bodies implicated in a variety of cellular processes (72, 73).

### KAP1 asymmetry is functional for recruiting binding partners

To understand if KAP1 structural asymmetry has functional implications when interacting with other binding partners, we explored the direct interaction between KAP1 FL and full-length HP1α (HP1α FL). Previous studies have identified a fragment of 15 residues within the central unstructured region of KAP1 as responsible for HP1 binding (residues 483–497, HP1BD, Fig 1A) (42, 47). Moreover, mutations of the HP1BD abolish the HP1 binding and significantly reduce the repression activity of KAP1 (47). This interaction, which is independent from PTMs, is mediated by the CSD of HP1 that binds as a dimer to the HP1BD. All previous studies explored the binding between the two proteins using purified fragments, specifically the CSD of HP1 and the HP1BD peptide of KAP1. Here, using full-length HP1 and KAP1, we determined their binding affinity by isothermal titration calorimetry (ITC, Fig 5A), observing an equilibrium dissociation constant ($K_d$) of 176 nM for the KAP1–HP1 complex, the value which is in line with previous measurements obtained using protein fragments (47) and which confirms the strong affinity between the two proteins.

Furthermore, we determined the molecular weight of the complex by SEC-MALS, exploring a range of concentrations from 10 to 127 μM for KAP1 FL and from 50 to 500 μM for HP1α FL, that is, covering complex stoichiometries ranging from 1:1 to 1:10 to saturate binding. The proteins elute as single symmetric peaks with retention volumes of 12 ml for KAP1 FL, 16 ml for HP1α FL, and 11.8 ml for the HP1–KAP1 complex (Fig 5B), and no concentration dependence in either elution volume or mass estimation was observed (Fig 5C). Thus, our measurements estimate a molecular weight of ~231 kDa for the complex, only compatible with a dimer of KAP1 FL (182 kDa) bound to one dimer of HP1α FL (45 kDa), for a 2:2 stoichiometry of the KAP1–HP1α complex. This unexpected stoichiometry is confirmed by the fit of the molar ratio in the ITC experiments to ~1 (N = 0.98) (Fig 5A). The expected stoichiometry 2:4, having two HP1α dimers bound to the two HP1BDs of the KAP1 dimer (Mw of ~276 kDa) was never observed. Based on this finding, we speculate that the reason for such unexpected coupling with HP1 has to be related with the asymmetric nature that we observed for KAP1 in solution, which affects the ability to expose the HP1BD and recruit efficiently HP1 molecules. The ensemble of KAP1 structures fitting the SAXS data has in fact only one HP1BD fully exposed for HP1 recruitment, whereas the second is mostly compacted around the RBCC domain, thus less accessible for productive binding (Figs 2C, 3E, and 5D).

The complexity of KAP1–HP1–mediated repression is further enhanced by the flexible conformations of multidomain proteins involved in protein–protein and protein–chromatin interactions, all timely modulated by PTMs. The study of such systems benefits from the combination of structural information and measurements of the interaction dynamics between components at the sm level. Total internal reflection fluorescence (smTIRF) imaging approaches can directly observe the chromatin interaction kinetics of individual chromatin binders (75, 76). We used this technique to determine how KAP1 alters the interaction dynamics of HP1α with chromatin fibers, trimethylated at lysine 9 on histone H3 (H3K9me3) to observe the behavior of the complex in the context of chromatin (Fig 6A). Here, H3K9me3 containing chromatin fibers (labeled with a fluorescent dye, Atto647N) are immobilized in a flow cell, and fluorescently labeled HP1α (carrying an Atto 532 dye) is injected, in the presence or absence of KAP1. Transient HP1α binding interactions on individual chromatin fibers are then directly observed using

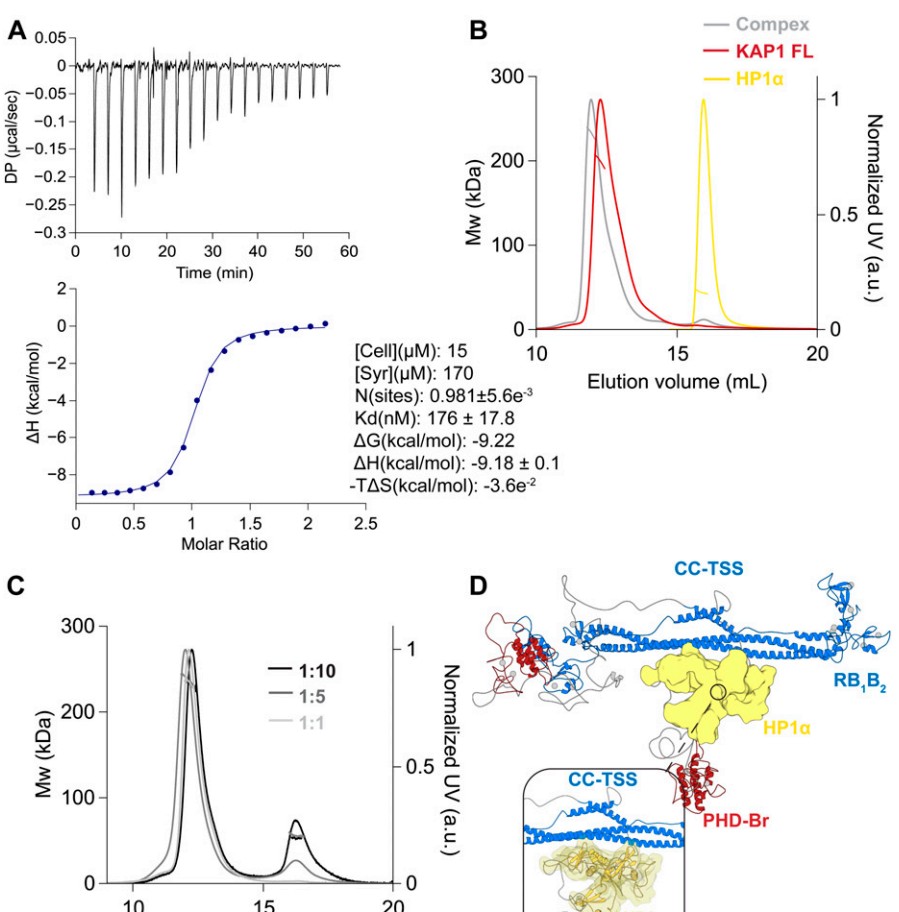

**Figure 5.   KAP1 dimer binds only one HP1 dimer.**
**(A)** ITC experiment titrating HP1α FL into KAP1 FL measures a tight interaction with a $K_d$ of 176 nM and a molar ratio of 0.98. **(B)** SEC-MALS analysis comparison between KAP1 FL (red), HP1α FL (yellow), and complex (grey). **(C)** SEC-MALS analysis of the complex at different concentration ratios. The estimated masses for the complexes eluting at 11.8 ml when mixtures contained KAP1:HP1 ratios of 1:1, 1:5, and 1:10, were 233 kDa, 239 kDa, and 232 kDa, respectively. The second peak eluting at 16 ml is the excess HP1 dimer with a mass of 45 kDa. **(D)** Representative model of the interaction between a KAP1 FL dimer (blue and red) and an HP1 FL dimer (yellow). The CSD of HP1 has been placed as to bind the accessible HP1BD (insert). Further details about the modeling of the HP1 FL dimer are reported in Fig S11.

smTIRF (Fig 6B and C). Compared with the short dwell times of individual HP1α molecules on chromatin (Fig 6B), the presence of 100 nM KAP1 resulted in a higher frequency of long (>4 s) binding events (Fig 6C). Analyzing dwell-time histograms (Fig 6D) revealed bi-exponential binding kinetics for HP1α (characterized by a fast and slow residence time $\tau_{off,1}$ and $\tau_{off,2}$, Table 1). It was previously shown that the slow exponential phase is attributed to bivalent chromatin binding by HP1α dimers (75). KAP1 addition indeed stabilized HP1α chromatin binding, exhibiting a larger percentage of bound molecules at times >1 s (Fig 6D). Conversely, binding events were less frequent in the presence of KAP1, as indicated in the slower binding kinetics (Fig 6E).

To more quantitatively elucidate the effect of KAP1 on HP1α binding dynamics, we tested the effect of KAP1 concentrations ranging from 50 to 400 nM. Whereas the fast binding times $\tau_{off,1}$ did not exhibit a systematic dependence on KAP1 concentration (Fig 6F), $\tau_{off,2}$ was increased by around threefold at KAP1 > 100 nM (Fig 6G). Thus, KAP1 stabilizes bivalent chromatin interactions of HP1α dimers, consistent with the fact that a KAP1 dimer can only bind one HP1 dimer. Conversely, binding rate constants (Fig 6H) were reduced by KAP1, consistent with slower diffusion dynamics of the HP1α-KAP1 complex. Finally, an analysis of the fluorescence intensities of the sm observations allowed us to quantify the

oligomeric state of HP1α in our experiments. In the absence of KAP1, HP1α was mainly monomeric (because of the low concentrations used) and only 8% dimers were observed (Fig 6I). Addition of 100 nM KAP1 resulted in an increase of dimers (up to 28%, Fig 6J), consistent with our previous observations that KAP1 binds and stabilizes a single HP1α dimer.

Thus, in the chromatin context represented by these conditions and using trimethylated nucleosomes, a KAP1 dimer is able to bind one dimer of HP1 with two binding sites for the histone tails packing the chromatin fiber. These bivalent interactions are longer lived than the interactions of HP1 alone, such that in the cell, this might translate into a more durable heterochromatin state that is able to spread along the DNA. It remains to be studied how the remaining PTMs and chromatin modifiers influence or are affected by this asymmetric architecture.

# Discussion

Gene expression in response to physiological and environmental stimuli is controlled by the cell at different levels by chromatin chemical modifications such that effector proteins can interpret

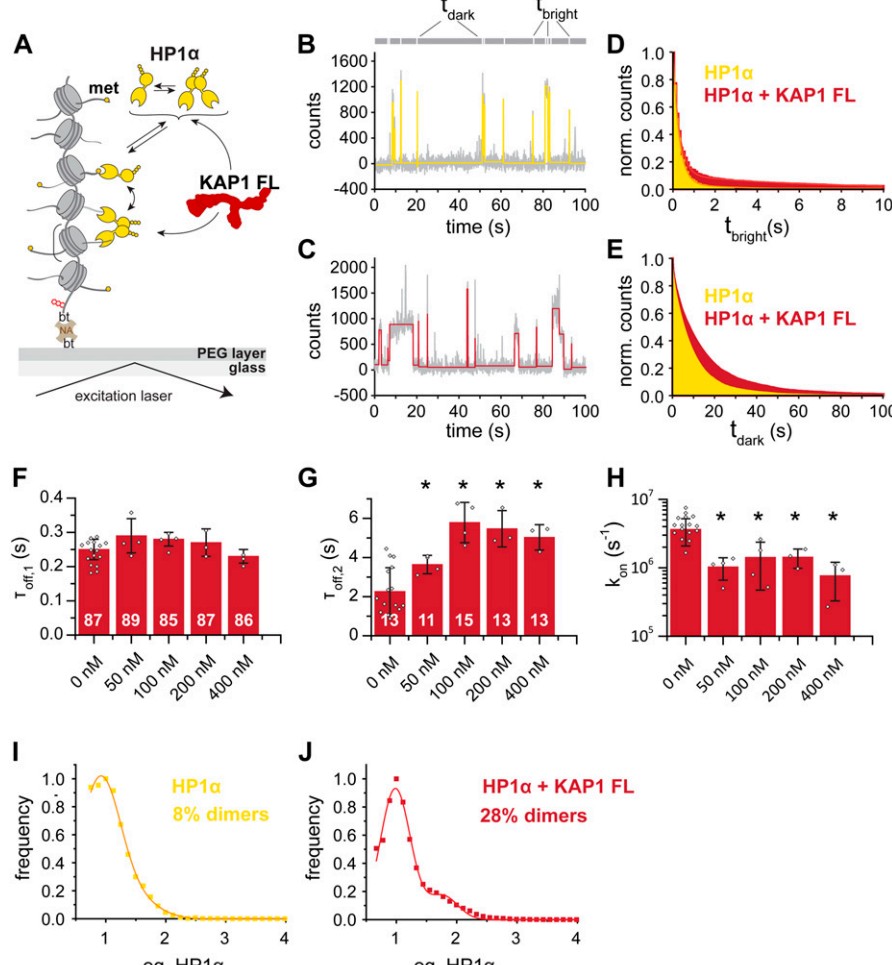

**Figure 6. KAP1 stabilizes HP1α–chromatin interactions.**

**(A)** Schematic representation of the smTIRF imaging experimental setup, showing HP1α interacting with chromatin fibers in the presence of KAP1. bt, biotin; NA, NeutrAvidin. **(B)** Characteristic fluorescence time trace (grey) of HP1α (3 nM) binding dynamics to a single chromatin fiber, in the absence of KAP1. High-fluorescence emission reveals the time an HP1α molecule is bound ($t_{bright}$), whereas low fluorescence emission indicates the absence of any bound molecules over a given time ($t_{dark}$). The trace is fitted with a step function (yellow). Each intensity peak represents one binding event. **(C)** Fluorescence time trace of HP1α–chromatin binding dynamics, in the presence of 100 nM KAP1. The trace is fitted with a step function (red). **(D)** Dissociation kinetics: normalized, cumulative histograms of HP1α dwell times in the absence of KAP1 (yellow) or presence of 100 nM KAP1 (red). Both histograms are fitted with a double-exponential function. For fit values, see Table 1. **(E)** Association kinetics: normalized cumulative histogram of times between binding events for HP1α alone (yellow) or HP1α in the presence of 100 nM KAP1 (red), fitted with mono-exponential function. For fit values, see Table 1. **(F, G)** Dwell times $\tau_{off,1}$ (F) and $\tau_{off,2}$ (G), demonstrating that KAP1 stabilizes HP1α–chromatin interactions. Numbers indicate % amplitude. Error bars: SD, n = 3–10 replicates, *$P < 0.05$ ($t$ test) versus 0 nM KAP1. For the fit values, see Table 1. **(H)** KAP1 reduces on-rates $k_{on}$. Error bars: SD, n = 3–10 replicates, *$P < 0.05$ ($t$ test) versus 0 nM KAP1. For the fit values, see Table 1. **(I)** A histogram of observed, normalized fluorescence peak intensities in kinetic traces reports on the oligomeric state of chromatin bound HP1α (yellow). This demonstrates that HP1α exists mostly as monomers at 3 nM concentration. Considering the observed labeling efficiency of 60%, the 8% of all observations correspond to dimers. **(J)** At 100 nM, KAP1 stabilizes HP1α dimers (28%) (red).

them and alter the chromatin state (77, 78, 79). These proteins are thought to have multiple roles and work in an integrated fashion, being modulated by PTMs themselves. KAP1 is a master regulator that exemplifies this level of complexity and coordination between PTMs and the binding of protein partners. KAP1 is not only SUMOylated but also phosphorylated at multiple sites (80) and the same applies to its binding partners, such as HP1 (50). The cross-talk between the PTMs within KAP1 and HP1 in fact affects their function in transcription control and DNA damage response (8), and

it is still not well understood at the molecular level. This is mainly due to the very flexible nature of these proteins, both containing multiple domains linked by disordered linkers, which so far have precluded the solution of their full-length structure. Therefore, we have used here an integrative modeling approach combining solution scattering data, flexible fitting, biophysical and biochemical analyses, as well as single-molecule experiments to reveal the molecular architecture of full-length KAP1 and found an intrinsic flexibility of its C-terminal domains, which generated an asymmetric

**Table 1. smTIRF values.**

| KAP1 (nM) | $\tau_{off,1}$ (s) | % $A_1$ | $\tau_{off,2}$ (s) | % $A_2$ | $k_{on}$ (M⁻¹s⁻¹) × 10⁶ | n |
|---|---|---|---|---|---|---|
| 0[a] | 0.25 ± 0.03 | 87 ± 7 | 2.26 ± 1.22 | 13 ± 7 | 3.64 ± 1.56 | 10 |
| 50 | 0.29 ± 0.05 | 89 ± 3 | 3.64 ± 0.47 | 11 ± 3 | 1.03 ± 0.37 | 4 |
| 100 | 0.28 ± 0.02 | 85 ± 3 | 5.79 ± 1.03 | 15 ± 3 | 1.42 ± 0.95 | 4 |
| 200 | 0.27 ± 0.04 | 87 ± 2 | 5.47 ± 0.93 | 13 ± 2 | 1.43 ± 0.45 | 3 |
| 400 | 0.23 ± 0.02 | 86 ± 11 | 5.03 ± 0.65 | 14 ± 11 | 0.77 ± 0.44 | 3 |

Fit results from smTIRF measurements of HP1α in the presence of the indicated concentrations of KAP1. N denotes the number of independent experiments, each contributing >100 kinetic traces.
[a]Values taken from reference (76).

arrangement of the PHD-Br domains in solution. On one hand, as other TRIM family members, KAP1 contains an N-terminal RBCC domain that consistently folds into a symmetric antiparallel long CC dimer, with the $RB_1B_2$ modules located at the opposite ends of it. On the other hand, the C-terminal domains are naturally more flexible as linked to the RBCC through very long, unstructured linkers. Flexible fitting to the SAXS scattering curves resulted in an ensemble of conformations featuring one PHD-Br domain closer to one $RB_1B_2$ domain, leaving the second more distant to the main core of KAP1, and thus more accessible (Fig 3). This solution is reminiscent of previous findings for the C-terminal PRYSPRY domains of TRIM25 (56), where although in crystallographic conditions, both domains were interacting along the CC region, in solution, SAXS showed transient interactions with the CC domain, hinting to a much more dynamic conformation for the C-termini in more physiological conditions (56, 81).

Is this asymmetric feature observed in solution associated with KAP1 function? On one hand, auto-SUMOylation assays have shown that the RING domain does play a role in the SUMOylation of the C terminus. As shown by us and others (39), SUMOylation happens at lysine residues at the C-terminal end and some SUMOylation sites remain SUMOylated after the PHD-Br domain is compromised with catalytic incompetent mutations (e.g., C651A). The remaining SUMOylation could be due to (i) the existence of another E3 ligase active site or (ii) the PHD being still active despite C651A mutation. Here, we provide evidence that the RING domain can act as an additional catalytic domain SUMOylating those sites or can act synergistically with the PHD domain for the same purpose. Unfortunately, KAP1-bearing RING truncations together with PHD-Br inactive mutations proved to be insoluble, not allowing us to definitively distinguish between these two situations. RING domains in other TRIM family members dimerize to preserve E3 ligase activity (e.g., TRIM32 (62), TRIM5α (59), or TRIM23 (82)), but the RING domain of KAP1 appears to be monomeric (29). This finding supports our model in suggesting a possible synergy between the RING and the PHD domain, which are in proximity based on our SAXS-based models, as well as putative SUMOylation sites at the C-terminal domains. Therefore, we conclude that KAP1 has two separate SUMO E3 ligase-active sites (31, 32, 39), which might work independently or synergistically depending on the substrate available.

One obvious reason for KAP1 asymmetry is to modulate KAP1 interactions with binding partners such as HP1 (Fig S11 and reference (83)). Previous data using the CSD and a small peptide from KAP1 showed that as expected, the CSD dimerized and bound one KAP1 peptide (47). According to our MALS data, this interaction is maintained when using FL proteins but involving only one of the two available HP1BDs of KAP1. This stoichiometry suggests that accessibility, in isolation and in the chromatin context, of the two available HP1 binding domain is not equal. As this binding asymmetry could be triggered upon HP1 binding, it can also be reasonable to think that the asymmetry of the C-terminal domains found in solution is responsible to modulate HP1 recruitment. Within chromatin fibers, KAP1 dimers recruit indeed HP1 as dimers to trimethylated nucleosomes such that two CDs per complex can reach for the histone tails. As this interaction is more durable with KAP1, we can speculate that in cells, the binding of KAP1 stabilizes HP1 dimerization in the chromatin facilitating heterochromatin formation and spreading. Regarding the mode of interaction, it is interesting to notice how more than half of the known KAP1 PTMs occur next to the HP1-binding domain. Although the HP1–KAP1 direct interaction is independent from PTMs, these may also play a role possibly affecting complex stability and chromatin interactions, as, for instance, in the case of S473 phosphorylation that has been shown to affect complex formation and modulate the expression of certain genes (52). It remains to be studied how such PTMs affect the asymmetric conformation of the C-terminal domains of KAP1. Similarly, this functional asymmetry in recruiting partners with controlled stoichiometry can be present for other KAP1 interacting partners engaged by the C-termini, such as SETDB1, or that interact with the CC domain, such as the melanoma antigen genes protein or KRAB-ZFPs (9, 26, 44). Related to this, while this article was under revision, it has been proposed that KAP1 FL binds only one KRAB domain at the center of the CC domain (33, 84).

What is then the ultimate purpose of KAP1 asymmetry? The answer can be linked to the multiple functions of KAP1, such that one RING/PHD-Br unit of the dimer is responsible for SUMOylation activity, whereas the other PHD-Br domain is SUMOylated and able to interact with histone modifiers. Both units could be independently modulated by PTMs or bound partners allowing for multiple layers of regulation and fast responses. The flexibility and intrinsically disordered nature of the linker regions of the KAP1 dimer would allow for this asymmetry, for an extended line of residues available for PTM regulation (85) and also for the rapid evolution of the microdomains within it (like the HP1BD), which could mutate to recruit different binding partners as needed without having to maintain a stable fold (86).

In conclusion, our findings shed light for the first time on the molecular architecture of full-length KAP1, its interaction with HP1 in the context of chromatin fibers, and the implications of its native flexibly and resulting asymmetry. Canonical structural biology techniques still struggle to provide a high-resolution characterization of systems of this kind, where multiple domains are connected by disordered linkers producing natively flexible complexes capable of plastic, dynamic interactions modulated by specific environmental conditions. Studying such dynamic systems using an integrative approach able to combine experimental data and molecular modeling results to be a powerful resource to understand their behavior in the cellular environment.

## Materials and Methods

### Expression and purification of recombinant proteins

Recombinant proteins were expressed and purified from *Escherichia coli*. The KAP1 variants DNA coding sequences were optimized for *E. coli* expression (Genscript) and cloned into the first multi-cloning site of the pETDuet-1 vector, preceded by a $His_6$ tag and a tobacco etch virus protease cleavage site. The plasmids were transformed in Rosetta (DE3) and Rosetta (DE3) pLys cells (Promega). Cells were grown to an optical density of 0.7–0.8 in Luria–Bertani media. Protein expression was induced by the addition of 1 mM IPTG and subsequent growth overnight at 20°C. Cell pellets

were resuspended in lysis buffer (20 mM Hepes, pH 7.5, 500 mM NaCl, 10% [vol/vol] glycerol, and 2 mM Tris(2-carboxyethyl)phosphine [TCEP], 5 mM imidazole, and cOmplete Protease Inhibitor Cocktail [Roche]) and then lysed using high-pressure homogenizer (Avestin Emulsiflex C3). The resulting suspensions were centrifuged (13,000 rpm for 35 min at 4°C) and the supernatants were applied to an HisTrap HP column (GE Healthcare) previously equilibrated with loading buffer (20 mM Hepes, pH 7.5, 10% [vol/vol] glycerol, and 500 mM NaCl). The proteins were eluted with a gradient over 40 column volumes of elution buffer (20 mM Hepes, pH 7.5, 500 mM NaCl, 10% [vol/vol] glycerol, and 500 mM imidazole). Subsequently, pure fractions were buffer-exchanged into final buffer (20 mM Hepes, pH 7.5, 500 mM NaCl, 10% [vol/vol] glycerol, and 2 mM TCEP) using a HiPrep Desalting column (GE Healthcare) and stored overnight at 4°C in final buffer. The proteins were additionally purified by SEC in final buffer, flash-frozen in liquid nitrogen, and stored at –20°C. HP1α was purified as previously described (76).

## SEC-MALS

The molecular weights of the constructs were determined by SEC-MALS detector. The mass measurements were performed on a Dionex UltiMate3000 HPLC system equipped with a three-angle miniDAWN TREOS static light scattering detector (Wyatt Technology). The sample volumes of 100 $\mu$l at a concentration of 40 $\mu$M were applied to a Superose 6 10/300 GL column (GE Healthcare) previously equilibrated with 20 mM Hepes, pH 7.5, 300 mM NaCl, and 2 mM TCEP at a flow rate of 0.5 ml/min. The data were analyzed using the ASTRA 6.1 software package (Wyatt Technology), using the refractive index of the buffer as a baseline and the refractive index increment for protein dn/dc = 0.185 ml/g.

## Analytical ultracentrifugation

Sedimentation velocity data of the N-terminal RBCC domain of KAP1 (0.5 mg/ml) were obtained using a Beckman Optima XL-I analytical ultracentrifuge (Beckman Coulter) with an absorbance optical system. EPON double-sector centerpieces containing 200 $\mu$l of protein solution and 300 $\mu$l of sample buffer were centrifuged at 55,000 rpm and 20°C and data acquired with a radial increment of 0.003 cm with no delay between scans. The sedimentation of the protein was monitored at 292 and 280 nm. Data analysis was performed with sedfit (87). Buffer viscosity (0.01002 cp), density (1.04 g/cm$^3$), and protein partial specific volumes (0.72 g/cm) were estimated using the program sednterp (88).

## SAXS data collection and analysis

To exclude sample aggregation, the proteins were analyzed by size-exclusion chromatography in line with small-angle X-Ray scattering (SEC-SAXS). The data were collected at the European Synchrotron Radiation Facility, beamline 29 (BM29) at a wavelength of 0.99 Å with a sample to detector distance of 2.867 m and a PILATUS 1M detector, covering a momentum transfer of 0.0025 < q > 0.6 Å$^{-1}$ [q = 4$\pi$sin

($\theta$)/$\lambda$]. Measurements were made at 18°C. The samples were applied to a Superose 6 10/300 GL column (GE Healthcare) at a concentration of 9–12 mg/ml and run at a flow rate of 0.75 ml/min in 20 mM Hepes, pH 7.5, 500 mM NaCl, 10% (vol/vol) glycerol, and 2 mM TCEP. During the elution, 2,160 scattering measurements were taken with 1-s time frames. The in-house software *BsxCuBE* (Biosaxs Customized Beamline Environment) connected to a data processing pipeline (EDNA) (89) was used to control the real-time data display (two dimensional and one dimensional) and to provide the first automatic data processing up to a preliminary ab initio model. SAXS data were analyzed using the ATSAS package version 2.8.3 (90) and ScÅtter (91). For each sample, using Chromixs (92), an elution profile was generated with the integrated intensities plotted versus the recorded frame number. Using Chromixs, 30 buffer frames were averaged and used to (i) subtract the buffer average from each frame of the sample peak selected and (ii) calculate the corresponding radius of gyration ($R_g$). The subtracted peak region was selected in Chromixs and averaged to generate the final scattering curve used for subsequent analysis. The scattering curves were initially viewed in *PRIMUS* (93) where the $R_g$ was obtained from the slope of the Guinier plot within the region defined by $q_{min} < q < q_{max}$, where $q_{max} < 1.3/R_g$ and $q_{min}$ is the lowest angle data point included by the program (Fig S3). When one dimension of a scattering particle is greater than the other two (e.g., a rod particle), "Rod" Guinier analysis in PRIMUS can be used to calculate the radius of gyration of its cross section, $R_{gc}$ (Fig S4). The $P(r)$ function, the distribution of the intra-atomic distances (r) in the particle, was generated using the indirect transform program *GNOM* (94). The maximum distance ($D_{max}$) was selected by letting the $P(r)$ curve decay smoothly to zero. As our molecules are rod-like and flexible, the $R_g$ was also estimated from the $P(r)$ function, such that, unlike the Guinier $R_g$ estimation, the $P(r)$ $R_g$ calculation takes the whole scattering curve into account. Ab initio models were produced using GASBOR (65) imposing a P1 symmetry with prolate (elongated protein) anisotropy until a $\chi^2$ of 1 was reached. DATPOROD, DATMOW, and DATVC within the ATSAS package were used to estimate the Porod volume (Vp) and the concentration-independent estimate of the MW for the proteins. The final figures were generated using VMD (95), PyMOL (Schrödinger, LLC), and Chimera (96).

## Molecular modeling and flexible SAXS data fitting

The preliminary KAP1 model was created using existing, homologous structures of the RING, B-box, and CC $\alpha$-helical domains, and the solved NMR structure of the PHD-Br domain (PDB: 2RO1) (36). Specifically, the SWISS Model server (97) was used to construct the RBCC domain using the following templates: the RING domain was based on the dimer of Rad18 (PDB: 2Y43) (98), the B-box 1 was based on the B-box domain of MuRF1 (PDB: 3DDT) (99), the B-box 2 was based on the B-box domain of TRIM54 (PDB: 3Q1D), and the CC domain was based on the CC of TRIM69 (PDB: 4NQJ) (61). To model the TSS domain in the central part of KAP1, we used the Robetta server (100), whereas MODELLER v9.14 (101) was used for the HP1-binding domain, assigning as a template the structure of EMSY protein in complex with HP1 (PDB: 2FMM) (102) and the structure of the small subunit of the mammalian mitoribosome (PDB: 5AJ3) (103). Finally, all the missing loops were modeled by Rosetta loop

modeling application v3.5 (104). In the process of generating starting models for fitting to SAXS data, each PHD-Br domain was randomly rotated and translated at a maximum distance of 140 Å (distance D in Fig 3B) from the RBCC domains using VMD (95). Afterwards, the linker region between each PHD-Br domain and its respective RBCC domain was modeled as a random coil using MODELLER (101). 1,000 models were generated and flexibly fitted to the SAXS data through a $x^2$–minimizing optimization procedure based on the nonlinear Cartesian NMA method called NOLB (68) and a novel SAXS profile calculator called Pepsi-SAXS (66). More precisely, for each initial model, we performed 100 optimization iterations. Each iteration comprised the computation of 60 slowest normal modes (using the NOLB tool), nonlinear structure deformation along these modes, and choosing the deformation with the least $x^2$ value to the experimental scattering profile (using Pepsi-SAXS). A steepest-descent minimization algorithm was used at the end of each iteration to keep the local topology (bonds and angles) in agreement with the initial structure. The flexible fitting method is available as a standalone executable called Pepsi-SAXS-NMA at (https://team.inria.fr/nano-d/software/pepsi-saxs/). The choice of 60 normal modes was based on the observation of high flexibility of the linker region. This number of modes allowed us to explore plenty of plausible conformations of the linker. The chosen slowest modes did not change the structure of the rigid domains, but only changed their relative orientation. ~60% of the initial models converged to low values of $x^2$ within the wall time assigned for fitting (24 h). The resulting structures produced a statistically relevant ensemble and were clustered using a new method tailored for clusters exhibiting Gaussian distributions, typical of structural ensembles. As for collective variables, the distance between the center of mass of the PHD-Br domain and that of the closest $RB_1B_2$ module was computed for each protomer. The models were classified according to these two distances and the cluster centers were extracted (Fig 3C and D).

### Negative stain electron microscopy

KAP1 Fl was diluted to 0.05 mg/ml in 20 mM Hepes, pH 7.5, 300 mM NaCl, and 2 mM TCEP and cross-linked with 0.1% glutaraldehyde for 2 h at 23°C. The reaction was stopped by addition of 100 mM Tris–HCl, pH 7.5. The sample was diluted 20 times, and adsorbed to a glow-discharged carbon-coated copper grid (EMS, Hatfield) washed with deionized water and stained with a solution of 2% uranyl acetate. The grids were observed using an F20 electron microscope (Thermo Fisher Scientific) operated at 200 kV. Digital images were collected using a direct detector camera Falcon III (Thermo Fisher Scientific) with 4,098 × 4,098 pixels. The magnification of work was 29,000× (px = 0.35 nm), using a defocus range of −1.5 to −2.5 $\mu$m. After manual picking of 400 particles, Relion (105) was used to sort them into 2D class averages.

### H3K9me3 synthesis

H3K9me3 was synthesized as previously described (76). In short, the peptide H3(1–14) K9me3-NHNH$_2$ (carrying a C-terminal hydrazide) was synthesized by solid phase peptide synthesis (SPPS). The truncated protein H3(Δ1–14)A15C was recombinantly expressed as

N-terminal fusion to SUMO, the N-terminal SUMO was cleaved by SUMO protease and the protein purified by RP-HPLC. For ligation, in a typical reaction, 3 $\mu$mol H3(1–14)K9me3-NHNH$_2$ was dissolved in ligation buffer (200 mM phosphate pH 3, 6 M GdmCl) at −10°C. NaNO$_2$ was added dropwise to a final concentration of 15 mM and incubated at −20°C for 20 min. H3(Δ1–14)A15C was dissolved in mercaptophenyl acetic acid (MPAA) ligation buffer (200 mM phosphate, pH 8, 6 M GdmCl, and 300 mM MPAA) and added to the peptide, followed by adjustment of the pH to 7.5. The ligation was left to proceed until completion (as observed by RP-HPLC). The product (H3K9me3A15C) was purified by semi-preparative RP-HPLC. For desulfurization, H3K9me3A15C was dissolved in TCEP desulfurization buffer (200 mM phosphate, pH 6.5, 6 M GdmCl, and 250 mM TCEP). Glutathione (40 mM) and a radical initiator, VA–044 (20 mM) were added, followed by a readjustment of the pH to 6.5. The reaction mixture was incubated at 42°C until completion (verified by RP-HPLC and ESI-MS). The final product, H3K9me3, was purified by semi-preparative RP-HPLC, lyophilized and kept at −20°C for further use. A final characterization of the modified histones was performed by analytical RP-HPLC and ESI-MS.

### Chromatin preparation

As previously described (76), chromatin arrays were reconstituted at a concentration of around 1 $\mu$M per mononucleosome, at a scale of 100 pmol. Array DNA (12 × 601 with 30 bp linker DNA) was mixed with 1 equivalent (per nucleosome positioning sequence) of recombinant human histone octamer, containing H3K9me3, in reconstitution buffer (10 mM Tris, pH 7.5, 10 mM KCl, and 0.1 mM EDTA) containing 2 M NaCl. 0.5 M equivalents of MMTV DNA was added to prevent oversaturation. In the case of reconstituted chromatin containing linker histone, 0.5, 1, or 1.5 equivalents H1.1 were also added to the DNA/octamer mixture. The reactions were gradually dialyzed from high salt buffer (10 mM Tris, pH 7.5, 1.4 M KCl, and 0.1 mM EDTA) to reconstitution buffer over 12 h with a two-channel peristaltic pump. After the dialysis the reconstituted chromatin arrays were analyzed by non-denaturing 5% PAGE in 0.5 × Tris–borate–EDTA running buffer or on a 0.6% agarose gel following ScaI restriction digest.

### HP1 labeling

HP1$\alpha$ was labeled as described in reference (76). A short tripeptide (Thz-G$_2$-C$_3$-CONH$_2$, Thz: thiazolidine) was synthesized by SPPS. 1 mg Atto 532-iodoacetamide (5 eq) was used to label the tripeptide in 200 mM phosphate pH 7.3, 5 M GdmCl. The reaction was followed by analytical-HPLC and MS, after completion quenched by addition of TCEP, purified by semi-preparative RP-HPLC, and lyophilized. The thiazolidine was opened by treatment with 2 M methoxylamine at pH 5. The labeled tripeptide was finally purified by semipreprative RP-HPLC, lyophilized, and kept at −20°C until further use. HP1$\alpha$ was expressed as a fusion to the $Npu^N$ split-intein at its C terminus, followed by a hexahistidine tag. Expression was induced in $E. coli$ BL21 DE3 cells with 0.25 mM IPTG overnight at 18°C. After cell lysis in lysis buffer (25 mM phosphate, pH 7.8, 50 mM NaCl, 5 mM imidazole, and 1x protease inhibitor/50 ml), HP1$\alpha$ was purified over Ni-affinity resin and eluted with elution buffer (25 mM phosphate, pH 7.8, 50

mM NaCl, and 400 mM imidazole). Eluted fractions were pooled and further purified by anion exchange chromatography, using a 1-ml HiTrap Q FF column and a gradient from low (25 mM phosphate, pH 7.8, and 50 mM NaCl) to high salt buffer (25 mM phosphate, pH 7.8, and 1 M NaCl). A total of 500 $\mu$l of the expressed HP1-Npu$^N$ fusion constructs at a concentration of 50–100 $\mu$M were applied to a column of 125 $\mu$l SulfoLink resin slurry, containing an immobilized Npu$^C$ peptide (106). After 5-min incubation, the column was drained, followed by washes with wash buffer (100 mM phosphate, pH 7.2, 1 mM EDTA, and 1 mM TCEP) containing high (500 mM NaCl), intermediate (300 mM NaCl), and low salt (150 mM NaCl). 1 mM of tripeptide in labeling buffer (100 mM phosphate, pH 7.8, 50 mM MPAA, 200 mM MESNA, 150 mM NaCl, 10 mM TCEP, and 1 mM EDTA) was added to the column and incubated for 16 h at room temperature. The column was drained and the eluate was collected. The column was further washed using elution buffer (100 mM phosphate, pH 7.2, 200 mM MESNA, 150 mM NaCl, 10 mM TCEP, and 1 mM EDTA). The eluted protein was finally purified by SEC with a Superdex S200 10/300 GL in gel filtration buffer (50 mM Tris, pH 7.2, 150 mM NaCl, and 1 mM DTT). Fractions containing purified labeled HP1 were pooled and concentrated, glycerol was added to 30% (vol/vol), and frozen aliquots were stored at −80°C. A final characterization of the purified and labeled HP1 proteins was done by analytical RP-HPLC and ESI-MS.

### Single-molecule assays

Single-molecule measurements were performed as previously reported (76). Glass coverslips (40 × 24 mm) and microscopy slides (76 × 26 mm) containing four drilled holes on each side were cleaned by sonication in 10% alconox, acetone, and ethanol with washing in miliQ H$_2$O between each step. The slides/coverslips were incubated for 1 h in a mixture of concentrated sulfuric acid to 30% hydrogen peroxide (3:1). The coverslips and slides were thoroughly washed with miliQ H$_2$O, sonicated in acetone for 15 min, and then submerged in acetone containing 2% (3-aminopropyl)triethoxysilane (APTES) for silanization. The slides and coverslips were dried with a nitrogen flow and strips of double-sided tape were sandwiched between a coverslip and a slide to create four channels. The glass coverslips were passivated with a solution of 100 mg/ml mPEG(5000)-succinimidyl carbonate containing 1% biotin-mPEG-succinimidyl carbonate for 3 h. The channels were subsequently washed with water and T50 buffer (10 mM Tris, 50 mM KCl). For chromatin immobilization, 0.2 mg/ml NeutrAvidin in T50 injected for 5 min, followed by extensive washes with T50 buffer. Then, 500 pM reconstituted chromatin arrays in T50 buffer were injected into the NeutrAvidin treated flow chamber for 5 min, followed by T50 washes and imaging buffer (50 mM Hepes, 130 mM KCl, 10% [vol/vol] glycerol, 2 mM 6-hydroxy-2,5,7,8-tetramethylchromane-2-carboxylic acid [Trolox], 0.005% Tween-20, 3.2% glucose, and glucose oxidase/catalase enzymatic oxygen removal system). Chromatin coverage was observed with a TIRF microscope (Nikon Ti-E) by fluorescent emission in the far-red channel upon excitation by a 640 nm laser (Coherent Obis). Dynamic experiments were initiated by influx of 3 nM Atto 532–labeled HP1, as well as the indicated KAP1 concentrations in imaging buffer. All smTIRF experiments were performed at room temperature (22°C). HP1 dynamics were observed with an EMCCD camera (Andor iXon) in

the yellow/orange channel using a 530-nm laser line for excitation at 20 W/cm$^2$. 10,000 frames were acquired at 20 Hz over a 25 × 50-$\mu$m observation area at a resolution of 160 nm/pixel. Every 200 frames, an image of the chromatin positions in the far-red channel was recorded for drift correction. For each chromatin fiber, an individual trace was extracted using a custom-made semi-automated MATLAB (MathWorks) script, as described in reference (76). After an initial baseline correction and a drift correction, a peak-finding algorithm was used to detect individual chromatin array positions. Fluorescence intensity traces for each chromatin position were obtained by integrating over a circle of 2-pixel radius. Individual HP1 fluorescence peaks were included based on point-spread-function and distance cutoffs. To ensure that only single molecules were analyzed, peaks exhibiting step-wise bleaching kinetics were excluded from the analysis. Kinetics were extracted from fluorescence traces using a semi-automated thresholding algorithm. Cumulative histograms were constructed from dark and bright intervals and fitted to mono or biexponential functions. For intensity analysis, normalized intensity histograms were constructed over several hundred kinetic traces.

### In vitro SUMOylation assay

The in vitro SUMOylation assay was conducted using a commercial kit from Abcam (ab139470) using 1 $\mu$M target protein. The mixtures (E1, E2 ubc9, SUMO1, ATP-Mg, and target proteins) were set up in 20 $\mu$l of 10× SUMO reaction buffer and incubated at 37°C for 2 h. The reactions were collected every 20 min to study the progression of the SUMOylation process. The reaction mixtures were subsequently separated using SDS–PAGE (4–12%) and were subjected to Western blot analysis using anti SUMO1 antibody. Gels were scanned with a Fusion FX 7 (Witec) and analyzed using ImageJ (74). Experiments were performed in triplicate (un-cropped gels are shown in Fig S7).

### ITC

ITC experiments were performed using a MicroCal PEAQ ITC from Malvern. KAP1 FL and HP1 FL were buffer-exchanged into 20 mM Hepes, pH 7.5, 300 mM NaCl, and 2 mM TCEP and concentrated to 15 and 170 $\mu$M, respectively. HP1 FL containing solution was injected into the KAP1 solution (2 $\mu$l of HP1 per injection at an interval of 180 s, a total of 19 injections into the cell volume of 300 $\mu$l, with stirring speed of 800 rpm, at 25°C). HP1 FL was also injected into buffer to determine the unspecific heat of dilution. However, subtracting this experimental heat of dilution was not sufficient to get a good fit, so the last injections were used to better estimate the heat of dilution and subtract a straight line from our data. The experimental data were fitted to a theoretical titration curve ("one set of sites" model) using software supplied by Microcal, with n (number of binding sites per monomer), $\Delta H$ (binding enthalpy, kcal/mol), and $K_d$ (dissociation constant, M) as adjustable parameters. Thermodynamic parameters were calculated from the Gibbs free energy equation, $\Delta G = \Delta H - T\Delta S = -RT \ln K_a$, where $K_a$ is the association constant, and $\Delta G$, $\Delta H$, and $\Delta S$ are the changes in free energy, enthalpy, and entropy of binding, respectively. T is the absolute temperature and R = 1.98 cal mol$^{-1}$ K$^{-1}$.

## Data Availability

SAXS data and models were deposited in the Small Angle Scattering Biological Data Bank SASBDB with accession codes SASDEV6, SASDER7, and SASDEW6. Other data are available from the corresponding authors upon request.

## Supplementary Information

## Acknowledgements

We thank the Ecole Polytechnique Fédérale de Lausanne (EPFL) Protein Production and Structure Core Facility for providing the equipment for the biophysical characterization of the protein complexes; Natacha Olieric and Bruno Correia for the use of their MALS machines; Cy Jeffries for his assistance with SAXS data preparation and analysis; Martin Moncrieffe for his help with AUC data analysis; Luciano Abriata, Kelvin Ka Ching Lau, and Florence Pojer for helpful discussions; BM29 staff scientists for their assistance in SAXS data collection; and EPFL for funding.

### Author Contributions

G Fonti: conceptualization, data curation, formal analysis, validation, investigation, visualization, methodology, and writing—original draft, review, and editing.
MJ Marcaida: conceptualization, resources, data curation, formal analysis, supervision, validation, investigation, visualization, methodology, and writing—original draft, review, and editing.
LC Bryan: resources, formal analysis, validation, investigation, and methodology.
S Träger: resources, data curation, software, formal analysis, visualization, and writing—review and editing.
AS Kalantzi: resources and formal analysis.
P-YJL Helleboid: resources and methodology.
D Demurtas: resources, data curation, formal analysis, validation, visualization, and methodology.
MD Tully: resources, formal analysis, and methodology.
S Grudinin: resources, data curation, software, formal analysis, visualization, and writing—review and editing.
D Trono: resources and formal analysis.
B Fierz: resources, formal analysis, validation, and writing—review and editing.
M Dal Peraro: conceptualization, data curation, formal analysis, supervision, funding acquisition, validation, investigation, visualization, methodology, project administration, and writing—original draft, review, and editing.

### Conflict of Interest Statement

The authors declare no conflict of interest.

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
