## [Reviewer comments · Life Science Alliance]

Life Science Alliance

KAP1 is an antiparallel dimer with a functional asymmetry

Giulia Fonti, Maria Marcaida, Louise Bryan, Sylvain Traeger, Alexandra Kalantzi, Pierre-Yves Helleboid, Davide Demurtas, Mark Tully, Sergei Grudinin, Didier Trono, Beat Fierz, and Matteo Dal Peraro

DOI: <https://doi.org/10.26508/lsa.201900349>

Corresponding author(s): Matteo Dal Peraro, Ecole Polytechnique Fédérale de Lausanne

Review Timeline:

Submission Date:	2019-02-18
Editorial Decision:	2019-03-18
Revision Received:	2019-06-06
Editorial Decision:	2019-07-02
Revision Received:	2019-08-03
Accepted:	2019-08-05

Scientific Editor: Andrea Leibfried

Transaction Report:

March 18, 2019

Re: Life Science Alliance manuscript #LSA-2019-00349-T

Matteo Dal Peraro
Ecole Polytechnique Fédérale de Lausanne
School of Life Sciences, Institute of Bioengineering
EPFL SV IBI LBM, AI1145, station 15
AAB 045
CH 1015, Lausanne
Switzerland

Dear Dr. Dal Peraro,

Thank you for submitting your manuscript entitled "KAP1 is an antiparallel dimer with a natively functional asymmetry" to Life Science Alliance. The manuscript was assessed by expert reviewers, whose comments are appended to this letter.

As you will see, the reviewers appreciate your work but think that more support for your conclusions is needed. They provide constructive input on how to provide such support, and we would thus like to invite you to provide a revised version of your work, addressing the individual criticisms raised by the reviewers. Further reaching insight is not needed, but as you will see, better support for the proposed asymmetry and stoichiometry is required (reviewer #1 and #2). The suggestions for improvements made by the technical reviewer #3 should get addressed, too.

Thank you for this interesting contribution to Life Science Alliance. We are looking forward to receiving your revised manuscript.

Sincerely,

B. MANUSCRIPT ORGANIZATION AND FORMATTING:

Reviewer #1 (Comments to the Authors (Required)):

KAP1 (also known as TIF1beta or TRIM28) performs a key role in regulating transcription, in

particular silencing, but there is not much information on its structural organization. In this manuscript, Fonti et al. integrates biochemical characterization, small angle X-ray scattering data, molecular modeling and single-molecule experiments to understand KAP1 structure and the functional implications for KAP1-HP1 mediated chromatin repression. The conclusion that KAP1 is an elongated antiparallel dimer like other TRIM proteins is not so surprising but is an important threshold issue that the authors convincingly establish here. On the other hand, the conclusion that KAP1 FL is asymmetric with respect to the PHD-Bromo domains rests on much weaker ground, which is unfortunate because the authors propose that the asymmetry of KAP1 has an important role in KAP1-HP1 mediated chromatin repression. Although this is intriguing, it rests on weaker ground and is not clearly explained. On the one hand, the authors seem to imply that KAP1 is inherently asymmetric and that this affects HP1 binding, but this relies only on weak data (modeling of SAXS curves). The authors do not consider the possibility that KAP1 is simply a flexible dimer wherein asymmetry is imposed by HP1 binding. Furthermore, how KAP1 asymmetry is mechanistically relevant to the overall process of chromatin repression is not very clear (at least not to this reviewer).

(1) The conclusion that KAP1 is an elongated antiparallel CC dimer similar to other TRIM proteins is very well supported by a collection of SEC-MALS, analytical centrifugation, SAXS, and EM data. The authors further conclude from modeling the SAXS data that full-length KAP1 is asymmetric with respect to how the PHD-Bromo domains are oriented with respect to the CC core. However, the authors do not make it explicitly clear whether the RBCC domain is also asymmetric. In Fig. 2c, the RBCC bead model appears asymmetric - was this calculated without imposition of 2-fold symmetry? Is this the only bead model that is consistent with the scattering data? What happens if 2-fold symmetry is imposed? It seems unlikely that the authors only obtained a single solution that is consistent with the raw SAX curve (Fig. 2e), and the authors need to show all the other solutions and discuss their significance.

(2) The authors model KAP1 RBCC as a symmetric (or closely symmetric) dimer, and suggest that the asymmetry of full-length KAP1 arises only from the differing dispositions of the PHD-Bromo domains relative to the CC dimer core. Again, they show that a single KAP1 model (Fig 3d) can account for the raw SAXS data (Fig 3e) while at the same time claiming a high degree of flexibility. This seems contradictory and the authors need to explain how they could obtain such a result. It is also important that high flexibility and asymmetric disposition of the PHD-Bromo domains are distinct aspects of the analysis - the SAXS data can only convincingly indicate the former. The authors need to come up with additional evidence for the latter.

(3) The authors seem to imply that the RING and PHD-Bromo domains specify SUMO different modification sites. If true, then this needs to be explored fully; otherwise the autSUMOylation experiments are not very well developed and its significance to the overall study is quite minimal.

(4) Experiments showing that KAP1 binds to HP1 at a stoichiometry of 2:2 (1 KAP1 dimer per HP1 dimer) seem convincing. The smTIRF experiments also convincingly show that the HP1 dwell times on immobilized chromatin fibers are shorter in the absence of KAP1, and much longer in the presence of KAP1. However, it is again not explained fully how the asymmetry of KAP1 is relevant here: is the 2:2 stoichiometry a consequence of the proposed asymmetry of KAP1, or because binding of 1 HP1 dimer to HP1BD prevents binding to the second, thereby imposing asymmetry? What happens when one of the PHD-Bromo domains or one of the HP1BP sites is removed from the dimer?

Other comments:

Figure 1a: Yellow bar in KAP diagram should be labeled.

Figure 2c, 2f, 3d, 3e: Authors should show a gallery of solutions and how imposition of 2-fold symmetry affects the results.

Reviewer #2 (Comments to the Authors (Required)):

This manuscript describes a structural and biochemical analysis of the heterochromatic scaffolding protein KAP1. Small-angle X-ray scattering combined with computer modeling demonstrates that full length KAP1 forms a dimer with flexibly tethered PHD-Bromo domains that show partial association with the structured core. The authors further show that auto-sumoylation attributed to the PHD domain also depends on the RING domain. To conclude, KAP1's interaction with HP1 is studied by ITC, SEC-MALS and single molecule fluorescence.

This work adds valuable information to the structural and biochemical understanding of KAP1. It establishes the RING and coiled-coil (RBCC) domains as the structured core of the KAP1 dimer. The SAXS data shows that the PHD-Br domains are flexibly connected, and modeling suggests that on average one of the two PHD-Br domains is found close to the RBCC core. The single molecule work further establishes convincingly that HP1 residence time increases in the presence of Kap1.

My main issue with this manuscript is a overinterpretation of the data. The authors favor a model with rather static asymmetry where one domain is associated with the core and the other free floating, even suggesting negative cooperativity. However, it is at least equally likely that the PHD-Br-core interaction is weak, non-specific and stochastic. If it were as static and specifically associated with the RING domain as postulated, then more than a single class should show a thicker end in the EM, as crosslinking will favor an association. While the auto-sumoylation and HP1 association provide interesting observations, they poorly support the structural claim of asymmetry. The fact that the RING domain is implicated in auto-sumoylation hardly provides evidence for association with the PHD-Br domain. For HP1 the evidence for 1:1 interaction is weak due to the ITC showing a molar ratio of higher than 1.5 and thereby contradicting the SEC-MALS data that suggests a 1:1 complex.

Provided the authors take care of the issues listed below and remove the misleading emphasis on asymmetry, I think it will be suitable for publication in Life Science Alliance.

Major concerns:

* Fig. 3: The plot in 3c shows that there are many models falling into a broad peak. In Fig. 3d the authors should show a representative ensemble of structures observed in this peak, not just one chosen structure.

* Fig. 4: The difference between full length and Δ RING is hard to judge due to large error bars. How was the activity calculated? It seems that signals of the Kap1-SUMO bands were normalized to the 120 min full length without loading/internal control for every lane. This could be the source of the big deviations. Also, the free SUMO/E2-SUMO signals are alarmingly constant considering that the signal on Kap1 changes from 0 to 100% (Fig. S7). This experiment has analysis and experimental quality issues that should be taken care of.

* Fig. 5a: ITC is of poor quality. For the fit the authors removed the first 4 injections, which is too

much since injection 2-4 look OK. Visually, the heat rates do not indicate as a pronounced sigmoidal curve as shown in the fit, and I'm suspecting that the sigmoidal shape will not hold up if signals 2-4 are included. For a reliable K_d these data need to be improved, it may be necessary to use an instrument with a bigger cell.

* Fig. 5a: The authors avoid talking about the molar ratio observed in the ITC experiment and do not show the fitted value for N . Judging from the fit the molar ratio is between 1.5 and 2. This suggests that 2 HP1 dimers bind a KAP1 dimer and contradicts the SEC-MALS data. This needs to be resolved.

* Fig. 6j: An expected distribution (gray curve) is shown and used as the argument for 1:1 complex formation. How did the authors come up with this calculated curve? In this rather complex system where HP1 oligomerisation status depends on Kap1 and where the experimental conditions are below K_d at an excess of 30:1 Kap1:HP1 it is hard to imagine how so much tetrameric HP1 should form. The Kap1 excess used here strongly biases the system towards a 1:1 complex.

Minor points:

* Page 18 first paragraph "While the fast binding times $T_{off,2}$ " should be $T_{off,1}$.

* In the Fig. 5a the K_d is labeled with μM , while it is nM in the legend and main text.

Reviewer #3 (Comments to the Authors (Required)):

I cannot comment on biological significance and importance of here characterized overall organization of the full-length Kap1. But I can clearly stated that here combine solution scattering data, integrative modeling and single-molecule experiments is an excellent example of integrative structural biology.

High quality SEC-SAXS data together with SEC-MALS and ITC experiment delivered solid validation of molecular stoichiometry of KAP1 and KAP1-Hp1 complex. Well performed SAXS based rigid body modeling elucidate atomistic models of fl-KAP1 which explain functional flexibility of the protein. The manuscript is well written and the figures have also good quality. In overall I could not find any technical issues in here presented work.

The details like a comparison of cross-sectional R_g , deep explanation of $P(r)$ functions and their comparison with model $P(r)$'s, PEPSI-SAXS conformers validation are just only a few highlights from here well performed SAXS, MALS, modeling, MD integrative structural study
Well done.

Minor comment:

Tables S1 have wrong q_{min} value.

Missing Guinier plot. In supplement Figure S3 is only cross sectional R_g .

ABSTRACT: "solution scattering diffraction data"
Please avoid diffraction in solution scattering

Response to the reviewers

We would like to thank all the reviewers for their constructive and positive feedback on our work. We provide below a point-by-point detailed response to all their concerns and point to the relative modifications in the revised manuscript, which are highlighted in red.

Response to Reviewer #1

KAP1 (also known as TIF1beta or TRIM28) performs a key role in regulating transcription, in particular silencing, but there is not much information on its structural organization. In this manuscript, Fonti et al. integrates biochemical characterization, small angle X-ray scattering data, molecular modeling and single-molecule experiments to understand KAP1 structure and the functional implications for KAP1-HP1 mediated chromatin repression. The conclusion that KAP1 is an elongated antiparallel dimer like other TRIM proteins is not so surprising but is an important threshold issue that the authors convincingly establish here. On the other hand, the conclusion that KAP1 FL is asymmetric with respect to the PHD-Bromo domains rests on much weaker ground, which is unfortunate because the authors propose that the asymmetry of KAP1 has an important role in KAP1-HP1 mediated chromatin repression. Although this is intriguing, it rests on weaker ground and is not clearly explained. On the one hand, the authors seem to imply that KAP1 is inherently asymmetric and that this affects HP1 binding, but this relies only on weak data (modeling of SAXS curves). The authors do not consider the possibility that KAP1 is simply a flexible dimer wherein asymmetry is imposed by HP1 binding. Furthermore, how KAP1 asymmetry is mechanistically relevant to the overall process of chromatin repression is not very clear (at least not to this reviewer).

While we answer below to each specific point raised by this reviewer, we want to comment here regarding this general concern and criticism about our work. As discussed below, we give now a clearer explanation of the flexible nature of KAP1. It was not our intention to present KAP1 as a static model. Its flexible nature was originally presented in our work, however we hope this key discussion is now clearer and more accessible in the revised manuscript. Although flexible, KAP1 appears to have some level of asymmetry that emerges already from the SAXS data as our further analyses point out. This asymmetry is then even more apparent upon binding with HP1 as only one HP1 dimer is recruited despite the availability of two putative binding sites. Following this main reasoning we have tuned down the discussion of these aspects, starting from the title.

(1) The conclusion that KAP1 is an elongated antiparallel CC dimer similar to other TRIM proteins is very well supported by a collection of SEC-MALS, analytical centrifugation, SAXS, and EM data. The authors further conclude from modeling the SAXS data that full-length KAP1 is asymmetric with respect to how the PHD-Bromo domains are oriented with respect to the CC core.

However, the authors do not make it explicitly clear whether the RBCC domain is also asymmetric. In Fig. 2c, the RBCC bead model appears asymmetric - was this calculated without imposition of 2-fold symmetry?

The RBCC bead models were calculated using GASBOR without imposing 2-fold symmetry. Thus, the RBCC model we produced and showed in **Fig 2C** is not strictly symmetric. However, we think that when compared with the asymmetry detected for the FL KAP1 (mainly due to the C-terminal domains) the asymmetry of RBCC is minor. In solution, the RBB domains can likely explore slightly different conformations but at the level of resolution accessible by SAXS these deviations can be seen as symmetric states.

Is this the only bead model that is consistent with the scattering data? What happens if 2-fold symmetry is imposed?

To answer this question, we further investigated the symmetry of RBCC. We generated 20 additional bead models with DAMMIF, we superimposed them with SUPCOMB, averaged them with DAMAVER and refined them with DAMMIN with the aim to compare the results with and without imposing 2-fold symmetry (**Figures R1 and R2**). We observed that both the bead models generated with and without 2-fold symmetry similarly fit our RBCC SAXS data. As previously anticipated, this observation leads to the conclusion that at this low resolution, even though the N-terminal tails and flexible loops will not be likely perfectly symmetric in solution, the RBCC of KAP1 can be considered an antiparallel dimer with a 2-fold symmetry axis in the middle of the CC such that the RBB modules are located at opposite ends and ~150 Å apart. The latter can find different conformation that can be not necessarily symmetric in solution but might be considered as symmetric at the low resolution accessible to SAXS. We added further discussion on this point at page 8 and 10 of the revised manuscript.

Figure R1: RBCC DAMMIF models generated imposing 2-fold symmetry

Mean value NSD: 1.734
Standard deviation of NSD: 0.091

Figure R2: RBCC DAMMIF models generated without imposing 2-fold symmetry

It seems unlikely that the authors only obtained a single solution that is consistent with the raw SAXS curve (Fig. 2e), and the authors need to show all the other solutions and discuss their significance.

We thank the reviewer for this observation, which gives us the opportunity to clarify this point. The atomic model of the RBCC domain shown in **Fig 2E** actually represented one of the possible solutions that can describe our SAXS data. The ensemble of solutions we obtained that optimally fit the data are however highly consistent/similar as the main differences are given by slight modifications of the reciprocal orientation of the Ring, B-box1 and B-box2 domains. As we leave a representative model in **Fig 2E** we have added a larger ensemble of models in **Figure S6** (i.e., different solutions that fit our data with χ^2 values between 0.8 and 1) to give a sense of the (limited) conformational variation of the ensemble and provided in **Fig 2F** the fit of all models to the raw SAXS data. We have further discussed this point in the main text of the revised manuscript at page 8.

(2) The authors model KAP1 RBCC as a symmetric (or closely symmetric) dimer, and suggest that the asymmetry of full-length KAP1 arises only from the differing dispositions of the PHD-Bromo domains relative to the CC dimer core. Again, they show that a single KAP1 model (Fig 3d) can account for the raw SAXS data (Fig 3e) while at the same time claiming a high degree of flexibility. This seems contradictory and the authors need to explain how they could obtain such a result.

As discussed above for RBCC, the conformation showed in the old **Fig 3D** is only a representative state of the solutions that fit the SAXS data for FL KAP1, in particular the centroid of the distribution in **Fig 3C**. Therefore, a larger ensemble of models that satisfies the SAXS data can be identified for any given χ^2 cutoff. For optimized models with χ^2 lower than 2, there are for instance ~300 models that all maintain a similar conformation of the C-termini. This means that the FL KAP1 structure in solution is not described by only one single model but by all the models that fit the SAXS data. Of this structural ensemble, we now show (i) a small subset in cartoon representation in **Fig S7B**, and (ii) a more extended (300 models) subset in **Fig S7C** (in simpler representation), while for the sake of clarity we have preferred to leave in **Fig 3E** of the main text the representative centroid model, which captures the main architectural features of the whole ensemble. We have also clarified this important point in the main text at page 9 and in the figure caption. This aspect is also shown in **Fig 3F**, where we report the fit of the calculated scattering profiles of the models belonging to the first density level of the cluster as red lines ($\chi^2 < 1$). In this revised figure, we have also shown a magnified inset to make clearer the point that we are presenting an ensemble of models.

It is also important that high flexibility and asymmetric disposition of the PHD-Bromo domains are distinct aspects of the analysis - the SAXS data can only convincingly indicate the former. The authors need to come up with additional evidence for the latter.

As the flexibility of FL KAP1 is convincingly demonstrated by SAXS, we tried to extract additional evidence for asymmetry by further analyzing the data. We followed the same procedure as for RBCC before and generated 20 additional bead models imposing or not 2-fold symmetry. In the case of the bead models generated without imposing any symmetry, we can clearly observe how the shape is well conserved among the different models that exhibit an NSD value of 1.43 ± 0.07 (**Fig R3**). When applying 2-fold symmetry, the models show a significantly broader range of conformations and a higher NSD value of 2.98 ± 0.64 (**Fig R4**), indicating that an asymmetric bead model generation better fits our FL KAP1 SAXS data.

Figure R3: FL DAMMIN models generated without imposing 2-fold symmetry

Figure R4: FL DAMMIN models generating imposing 2-fold symmetry

To further elaborate on this point and better support the asymmetry of KAP1, the two mirror clusters obtained by NMA (shown in **Fig 3C**) were superimposed (**Fig 7SA**) highlighting that every model contains one PHD-Br domain at a distance d_{close} of the RB₁B₂ domain and another at a larger distance d_{far} . Similarly, in the new **Fig 3D**, we report a 1D representation of those PHD-Br domains plotted according to their distance to the RB₁B₂ domain. As expected, their distribution is not random, but they are organized in two separated Gaussian distributions centered at d_{close} (~60 Å) and d_{far} (~120 Å).

The density level containing 300 models close to the cluster center or the peak of the Gaussian distribution and highlighted as a solid black line in **Fig S7A** has been selected and further analyzed. The models belonging to this density level have been extracted and superimposed to recover common asymmetric features. In **Fig S7C** we present the ensemble of these 300 models in which we highlight the PHD-Br domain close to the RB₁B₂ module in red and the PHD-Br domain far from the module RB₁B₂ in blue. This representation shows a clear asymmetric density distribution of the two C-terminal domains that is common for all the models (one far and one close to the RB₁B₂ domains). Furthermore we selected models from the first density level, containing 10% of the cluster models and in **Fig S7B** we present the centroid and some representative models. Again we highlighted in red and in blue the C-terminal domains close and far to the RB₁B₂ module.

Thus, the asymmetry of the bead models revealed without imposing 2-fold symmetry is most likely dictated by this different arrangement of the PH-Br domains, as the RBCC domain can be considered symmetric at this level of resolution (see discussion on point 1).

As suggested by the reviewer, we have now more clearly separated the discussion on flexibility and asymmetry in the revised text at pages 9 and 10, as supported by these further analyses. We have also reduced the emphasis on asymmetry as emerged from the SAXS data and instead used the results for KAP1-HP1 binding (i.e., SEC-MALS and ITC) to further corroborate the asymmetry of KAP1 when in complex with other protein factors.

(3) The authors seem to imply that the RING and PHD-Bromo domains specify SUMO different modification sites. If true, then this needs to be explored fully; otherwise the autosumoylation experiments are not very well developed and its significance to the overall study is quite minimal.

We understand the concern of this reviewer and in fact a further and more complete investigation of the implications of the RING function on the different SUMO sites is undergoing in our laboratory. However, this was not the main focus of this work and the newly discovered properties of the RING domain are here only developed with respect to the global architecture proposed for the FL KAP1 system. We made this aspect clearer in the revised version at page 12. In particular, our SAXS analysis indicates that at least one PH-Br domain and neighboring SUMO sites are more compacted along the RBCC domain, thus in proximity to the RING domain. As several studies indicate that the RING domain acts as a E3 SUMO ligase, we wanted here to find out if our FL KAP1 model was compatible with the RING having a role in the auto-SUMOylation of the C-terminus of KAP1. Our results, although not precisely mapping the specific site involved in auto-SUMOylation are confirming this role for the RING domain, thus further corroborating the FL KAP1 models obtained by SAXS and integrative modeling.

(4) Experiments showing that KAP1 binds to HP1 at a stoichiometry of 2:2 (1 KAP1 dimer per HP1 dimer) seem convincing. The smTIRF experiments also convincingly show that the HP1 dwell times on immobilized chromatin fibers are shorter in the absence of KAP1, and much longer in the presence of KAP1. However, it is again not explained fully how the asymmetry of KAP1 is relevant here: is the 2:2 stoichiometry a consequence of the proposed asymmetry of KAP1, or because binding of 1 HP1 dimer to HP1BD prevents binding to the second, thereby

imposing asymmetry? What happens when one of the PHD-Bromo domains or one of the HP1BP sites is removed from the dimer?

As discussed for point 2, SAXS data appear to support some asymmetry of the C-terminal domains of FL KAP1 in solution, which directly effects the conformation of the HP1BD. Based on this observation we tend to think that HP1 asymmetric stoichiometry is eventually the effect and not the cause of this conformation. Given the proposed architecture of FL KAP1, we think that KAP1 hetero-dimers with only one PHD-Br domain or HP1BD would still be able to bind HP1.

Other comments:

Figure 1a: Yellow bar in KAP diagram should be labeled.

This point has been addressed in the new version of **Fig 1**.

Figure 2c, 2f, 3d, 3e: Authors should show a gallery of solutions and how imposition of 2-fold symmetry affects the results.

We have added **Figs R1-4** to address questions regarding the *ab initio* modeling imposing 2-fold symmetry and we have discussed this point in the main text at pages 9 and 10. We feel the discussion in the text could be sufficient but we are open to add **Figs R1-4** to the supplementary material. Moreover, **Figs 2** and **3** have been modified. The centroid structures in **Fig 2E** and **3E** have not been modified for the sake of clarity but have been complemented with the new **Figures S6** and **S7** showing the structural ensembles and well as the fit of the ensemble models to the SAXS data in **Fig 2F** and **3F**.

Response to Reviewer #2

This manuscript describes a structural and biochemical analysis of the heterochromatic scaffolding protein KAP1. Small-angle X-ray scattering combined with computer modeling demonstrates that full length KAP1 forms a dimer with flexibly tethered PHD-Bromo domains that show partial association with the structured core. The authors further show that auto-sumoylation attributed to the PHD domain also depends on the RING domain. To conclude, KAP1's interaction with HP1 is studied by ITC, SEC-MALS and single molecule fluorescence.

This work adds valuable information to the structural and biochemical understanding of KAP1. It establishes the RING and coiled-coil (RBCC) domains as the structured core of the KAP1 dimer. The SAXS data shows that the PHD-Br domains are flexibly connected, and modeling suggests that on average one of the two PHD-Br domains is found close to the RBCC core. The single molecule work further establishes convincingly that HP1 residence time increases in the presence of Kap1.

My main issue with this manuscript is a overinterpretation of the data. The authors favor a model with rather static asymmetry where one domain is associated with the core and the other free floating, even suggesting negative cooperativity. However, it is at least equally likely that the PHD-Br-core interaction is weak, non-specific and stochastic. If it were as static and specifically associated with the RING domain as postulated, then more than a single class should show a thicker end in the EM, as crosslinking will favor an association. While the auto-sumoylation and HP1 association provide interesting observations, they poorly support the structural claim of asymmetry. The fact that the RING domain is implicated in auto-sumoylation hardly provides evidence for association with the PHD-Br domain. For HP1 the evidence for 1:1 interaction is weak due to the ITC showing a molar ratio of higher than 1.5 and thereby contradicting the SEC-MALS data that suggests a 1:1 complex.

Provided the authors take care of the issues listed below and remove the misleading emphasis on asymmetry, I think it will be suitable for publication in Life Science Alliance.

We took the concerns of this reviewer in great consideration and, as also asked by Reviewer #1 (also see response above), we now better clarify the flexibility and asymmetry aspects that emerge from our analyses. This further discussion is added at page 9 and 10 of the revised manuscript and revised and new figures are added to the revision (i.e., **Figures R1-4** and new panels in **Figs 2 and 3**). We also followed the suggestions of this reviewer and revised the manuscript in order to tune down the overinterpretation of the asymmetry as discussed below in the point-by-point response to the specific major concerns.

Major concerns:

** Fig. 3: The plot in 3c shows that there are many models falling into a broad peak. In Fig. 3d the authors should show a representative ensemble of structures observed in this peak, not just one chosen structure.*

We thank the reviewer for having raised this point, we acknowledge that this was not clear enough in the original version as also Reviewer #1 was puzzled by the same aspect (see also previous response). We now make clearer that the models presented in the main text are only representative conformations of a broader ensemble of models that fit optimally well the SAXS data. We are not at all proposing a static conformation of KAP1, but rather a very flexible ensemble of conformation as originally expected for this multidomain system. Nonetheless, this intrinsic flexibility (see also response to Reviewer #1, point 2) seems to eventually produce an asymmetry in the conformation of the C-terminal domains, as emerged from SAXS, which can be now more visually and quantitatively recognized in the new **Fig 3** and **Fig S7**, where we

report additional information of the whole ensemble (see also response to reviewer #1, point 2 for further details). For sake of clarity we have preferred to leave only one representative model in the revised main text, which is able to summarize the architectural properties of the whole ensemble, but we report the fit to the raw SAXS data of the ensemble in **Fig 3F** and report more details of the whole ensemble in **Fig S7**.

** Fig. 4: The difference between full length and Δ RING is hard to judge due to large error bars. How was the activity calculated? It seems that signals of the Kap1-SUMO bands were normalized to the 120 min full length without loading/internal control for every lane. This could be the source of the big deviations. Also, the free SUMO/E2-SUMO signals are alarmingly constant considering that the signal on Kap1 changes from 0 to 100% (Fig. S7). This experiment has analysis and experimental quality issues that should be taken care of.*

We acknowledge these concerns and we have tried to improve the quality of our experiments. First, the bands were initially normalized to the 120 min FL lane producing a big value in the standard deviation. We are working with purified proteins and for this reason our internal control is represented by the SDS page gel, demonstrating how the two proteins have been equally loaded meaning that the difference in activity is not due to protein concentration. To clarify this point we re-analyzed the data previously collected and normalized using all the data for each experiment (i.e., 100% intensity is the sum of all the bands). In the revised **Fig 4** of the main text we now plot the mean value of each data point for the 3 repetitions with the standard error of the mean.

** Fig. 5a: ITC is of poor quality. For the fit the authors removed the first 4 injections, which is too much since injection 2-4 look OK. Visually, the heat rates do not indicate a pronounced sigmoidal curve as shown in the fit, and I'm suspecting that the sigmoidal shape will not hold up if signals 2-4 are included. For a reliable K_d these data need to be improved, it may be necessary to use an instrument with a bigger cell.*

** Fig. 5a: The authors avoid talking about the molar ratio observed in the ITC experiment and do not show the fitted value for N . Judging from the fit the molar ratio is between 1.5 and 2. This suggests that 2 HP1 dimers bind a KAP1 dimer and contradicts the SEC-MALS data. This needs to be resolved.*

To address the concerns about the quality of the ITC experiments and the implication for HP1-KAP1 stoichiometry and KAP1 asymmetry (which is in our opinion the crucial point of criticism of this reviewer), we performed a new ITC run titrating HP1 (170 μ M) into KAP1 (15 μ M). We performed 19 injections of 2 μ L each (except the first one that was 0.4 μ L) separated by an interval of 180 sec. In the analysis we included all the data points except the first injection. We obtained a K_d value of 176 nM, (which is very similar to the previous value of 162 nM) and an N value of 0.98 confirming a stoichiometry in which one HP1 dimer is bound to one KAP1 dimer (see revised **Fig 5** of the main text). These new ITC data are now fully consistent with SEC-MALS results, pointing to the same KAP1-HP1 stoichiometry. We hope that the new results and analysis will meet the quality standards expected by the reviewer. As discussed also in our response to Reviewer #1 (point 2), this result more robustly corroborates the asymmetry of the FL KAP1 upon binding, which was already hinted for the unbound protein by analysis of the SAXS data and it is at last confirmed by SEC-MALS and ITC when KAP1 is in complex with HP1.

** Fig. 6j: An expected distribution (gray curve) is shown and used as the argument for 1:1 complex formation. How did the authors come up with this calculated curve? In this rather complex system where HP1 oligomerisation status depends on Kap1 and where the experimental conditions are below K_d at an excess of 30:1 Kap1:HP1 it is hard to imagine how so much tetrameric HP1 should form. The Kap1 excess used here strongly biases the system towards a 1:1 complex.*

We thank the reviewer for the thoughtful comment, and agree fully with her/his assessment that under our measurement conditions the probability of finding HP1 complexes above a 2:1 (HP1:KAP1) stoichiometry is highly unlikely. Thus, from these experiments alone, the absence of 4:1 complexes cannot be concluded. Conversely, our SEC-MALS experiments, as reported in **Fig 5** and performed at appropriate molar KAP1/HP1 ratios, clearly demonstrate the 2:1 stoichiometry, as well as the new ITC experiments and analysis. We have thus removed the grey curve in **Fig 6J** and changed the subsequent discussion at page 18 of the revised manuscript to state that: “Addition of 100 nM KAP1 resulted in an increase of dimers (up to 28%, **Fig 6J**), consistent with our previous observations that KAP1 binds and stabilizes a single HP1a dimer.”

Minor points:

* Page 18 first paragraph "While the fast binding times $T_{off,2}$ " should be $T_{off,1}$.

* In the Fig. 5a the K_d is labeled with μM , while it is nM in the legend and main text.

All these minor points have been corrected.

Response to Reviewer #3

I cannot comment on biological significance and importance of here characterized overall organization of the full-length Kap1. But I can clearly stated that here combine solution scattering data, integrative modeling and single-molecule experiments is an excellent example of integrative structural biology.

High quality SEC-SAXS data together with SEC-MALS and ITC experiment delivered solid validation of molecular stechiometry of KAP1 and KAP1-Hp1 complex.

Well performed SAXS based rigid body modeling elucidate atomistic models of fl-KAP1 witch explain functional flexibility of the protein. The manuscript is well written and the figures have also good quality. In overall I could not find any technical issues in here presented work. The details like a comparison of cross-sectional Rg, deep explanation of P(r) functions and their comparison with model P(r)'s, PEPSI-SAXS conformers validation are just only a few highlights from here well performed SAXS, MALS, modeling, MD integrative structural study Well done.

We thank this reviewer for the extremely positive assessment of our work.

Minor comment:

Tables S1 have wrong qmin value.

We substituted with: q-range 0.0025 -0.5 Å⁻¹

Missing Guinier plot. In supplement Figure S3 is only cross sectional Rg.

We have added a new Fig S3 with the Guinier fitting.

ABSTRACT: "solution scattering diffraction data" Please avoid diffraction in solution scattering

Right: We have corrected this sentence.

July 2, 2019

RE: Life Science Alliance Manuscript #LSA-2019-00349-TR

Prof. Matteo Dal Peraro
Ecole Polytechnique Fédérale de Lausanne
School of Life Sciences, Institute of Bioengineering
EPFL SV IBI LBM, AAB048, station 19
AAB 048
CH 1015, Lausanne 1015
Switzerland

Dear Dr. Dal Peraro,

Thank you for submitting your revised manuscript entitled "KAP1 is an antiparallel dimer with a functional asymmetry". As you will see, the reviewers appreciate the changes introduced in revision and now support publication, pending some minor adjustments. We would thus be happy to publish your paper in Life Science Alliance pending final revisions necessary to address the remaining reviewer comments and to meet our formatting guidelines:

- please address the remaining comments of reviewer #2
- please add panel descriptions in the legend for Fig S4, S8 and S9, please add panel labels in Figures S10
- you display the same figure panel twice in S2D and 3C, please remove one
- please upload the manuscript text as a word docx file
- please upload all figure files as individual files (main and suppl figures) and without legends; the S figure legends and references should get incorporated into the main manuscript docx file
- please list 10 authors et al in your reference list
- please add a summary blurb in our submission system
- please check the author order carefully in our submission system
- all corresponding authors should link their profile in our submission system to their ORCID iDs, you should have received an email with instructions on how to do so

A. FINAL FILES:

B. MANUSCRIPT ORGANIZATION AND FORMATTING:

Sincerely,

Andrea Leibfried, PhD
Executive Editor

Life Science Alliance
Meyerhofstr. 1
69117 Heidelberg, Germany
t +49 6221 8891 502
e a.leibfried@life-science-alliance.org
www.life-science-alliance.org

Reviewer #1 (Comments to the Authors (Required)):

The authors have addressed my comments and critiques appropriately. I'm still not fully convinced that the scattering data are sufficient with regards to the asymmetry, but the authors make a convincing enough argument and discuss this issue much more clearly - with results now better delineated vs interpretation. This paper would be a good basis for further studies.

Reviewer #2 (Comments to the Authors (Required)):

The revised manuscript of Fonti et al. is significantly improved and is in my view acceptable for publication.

There are still a few minor issues:

Fig. S1: The AUC fit is poor, since the residuals are split up into two populations and are much bigger than the noise of the data. This should be mended with a better choice of parameters in sedfit.

Fig. 5A: The integrated heat data look too smooth compared with the relatively noisy raw data. Has the baseline been overoptimized? The fit is otherwise OK.

p. 8 line 5 should specify what the 280A refers to (length?).

p. 12: "As expected, the RBCC domain" should also refer to Fig. S10.

Reviewer #3 (Comments to the Authors (Required)):

I thank the authors for their thoughtful and thorough responses.

I commend the authors on doing an excellent job in revising the manuscript, which is now significantly improved. The manuscript is excellent and should serve as an example of how to derive and validate hypothesis relevant to the solution state.

Reviewer #2:

- we refitted our AUC data as asked by the reviewer and provided a new Figure S1;
- regarding the integrated heat data, we tried different baselines, however the data did not change significantly;
- all the other minor points have been corrected in the final version of the manuscript.

Editorial comments:

- we corrected all the supplementary figures and captions;
- the data in figures S2D and 3C look the same but they are not. One refers to the FL KAP1 and the other to the Δ KAP1. The systems are almost identical but different;
- all the rest has been hopefully addressed in the final submitted files.

August 5, 2019

RE: Life Science Alliance Manuscript #LSA-2019-00349-TRR

Prof. Matteo Dal Peraro
Ecole Polytechnique Fédérale de Lausanne
School of Life Sciences, Institute of Bioengineering
EPFL SV IBI LBM, AAB048, station 19
AAB 048
CH 1015, Lausanne 1015
Switzerland

Dear Dr. Dal Peraro,

Thank you for submitting your Research Article entitled "KAP1 is an antiparallel dimer with a functional asymmetry". It is a pleasure to let you know that your manuscript is now accepted for publication in Life Science Alliance. Congratulations on this interesting work.

DISTRIBUTION OF MATERIALS:

Again, congratulations on a very nice paper. I hope you found the review process to be constructive and are pleased with how the manuscript was handled editorially. We look forward to future exciting submissions from your lab.

Sincerely,

Andrea Leibfried, PhD
Executive Editor
Life Science Alliance
Meyershofstr. 1
69117 Heidelberg, Germany
t +49 6221 8891 502
e a.leibfried@life-science-alliance.org
www.life-science-alliance.org